



# Identifying Sensitivities in Flood Frequency Analyses using a Stochastic Hydrologic Modeling System

Andrew J. Newman[1], Amanda G. Stone[2], Manabendra Saharia[1,^], Kathleen D. Holman[2], Nans Addor[3], and Martyn P. Clark[1*]

[1]Research Applications Laboratory, National Center for Atmospheric Research, Boulder CO, 80503, USA
[2]Technical Service Center, Bureau of Reclamation, Lakewood CO, 80215, USA
[3]Geography, College of Life and Environmental Sciences, University of Exeter, Exeter, EX4 4RJ, UK
[^]now at: Department of Civil Engineering, Indian Institute of Technology, New Delhi 110016, India
[*]now at: University of Saskatchewan Coldwater Lab, Canmore, Alberta T1W 3G1, Canada, and Centre for Hydrology,
University of Saskatchewan, Saskatoon, Saskatchewan S7N 1K2, Canada

*Correspondence to*: Andrew J. Newman (anewman@ucar.edu)

**Abstract.** This study assesses sources of variance in stochastic hydrologic modelling to support flood frequency analyses. The major components of the modelling chain, including model structure, model parameter estimation, initial conditions, and precipitation inputs were examined across return periods from 2 to 100,000 years at two watersheds representing different hydro-climates across the western United States. Ten hydrologic model structures were configured, calibrated and run within the Framework for Understanding Structural Errors (FUSE) modular modelling framework for each of the two watersheds. Model parameters and initial conditions were derived from long-term calibrated simulations using a 100-member historical meteorology ensemble. A stochastic event-based hydrologic modelling workflow was developed using the calibrated models; millions of flood event simulations were performed at each basin. The analysis of variance method was then used to quantify the relative contributions of model structure, model parameters, initial conditions, and precipitation inputs to flood magnitudes for different return periods. The attribution of the variance of flood frequencies to each component of a stochastic hydrological modelling framework, including several hydrological model structures, is a novel contribution to the flood modelling literature. Results demonstrate that different components of the modelling chain have different sensitivities for different return periods. Precipitation inputs contribute most to the variance of rare events, while initial conditions are most influential for the more frequent events. However, the hydrological model structure and structure-parameter interactions together play an equally important role in specific cases, depending on the basin characteristics and type of flood metric of interest. This study highlights the importance of critically assessing model underpinnings, understanding flood generation processes, and selecting appropriate hydrological models that are consistent with our understanding of flood generation processes.

## 1 Introduction

Understanding flood risk is important to support infrastructure design and operations. Hydrologic hazard curves and flood hydrographs are used to evaluate hydrologic risks for a given facility (e.g., a dam). A hydrologic hazard curve is a curve that



relates probability of occurrence to magnitude of a flood. There are numerous approaches to developing these curves, including (1) statistical stream gauge analysis, e.g., calculating the annual exceedance probability (AEP) (National Research Council 1988); (2) 'design storm' rainfall-runoff hydrologic model estimates, where the return period of the flood is equal to the return

period of the precipitation (e.g. Packman and Kidd 1980; Boughton and Droop 2003; Reclamation 2006; Wright et al. 2020); (3) more complex fully stochastic rainfall-runoff modelling to explicitly represent the impacts of hydrological processes on floods (Rahman et al. 2002; Schaefer and Barker 2002; Nathan et al. 2003; Wright et al. 2014); and (4) analysis of paleoflood records (England et al. 2010). Typically, multiple methods are employed in these analyses to help understand the uncertainty of model results (e.g., England et al. 2014). Many of these methods rely on the assumption of AEP-neutrality, i.e., that a rainfall

event has a similar AEP to the flood event.

The assumption of AEP neutrality is often not verifiable or violated (e.g. Rahman et al. 2002; Kuczera et al. 2006; Small et al 2006; Pathiraja et al. 2012; Paquet et al. 2013; Ivancic and Shaw 2015; Sharma et al. 2018). One avenue to address this is to perform stochastic rainfall-runoff modelling. In stochastic rainfall-runoff modelling, flood frequency (FF) estimates are

typically produced using stochastic event simulations using a single hydrologic model with randomly perturbed model parameters, initial conditions (ICs), and precipitation event forcing scenarios from defined precipitation frequency distributions (Rahman et al. 2002; Paquet et al. 2013; Wright et al. 2020). This modelling chain permits deviations from AEP-neutrality and quantifies the impacts of IC, model parameter, and precipitation event forcing variability in FF estimates.

However, past research on hydrologic model behaviour also emphasises the differences in model performance and responses for various event types given different model parameters and structures across hydroclimates (e.g. Clark et al. 2008; Mendoza et al. 2015; Markstrom et al. 2016; Newman et al. 2017; Mizukami et al. 2019), highlighting the possible need to include multiple model structures in stochastic flood modelling studies. Model structure can vary widely. For example, a model may simply be defined by a loss methodology where an initial and continuous losses are defined at the start of and during the event

simulation (e.g. Boughton and Droop 2003), or can be more complex employing various methods to explicitly simulate the dominant hydrological processes (e.g., snow melt, surface runoff generation). Additionally, most methods used to perturb model parameters and meteorological forcings do not allow us to identify which components are the most sensitive in an FF estimate. Therefore, we systematically explored the sensitivity FF estimates to provide a better understanding of which components of the modelling chain have the most impact on FF estimates across example hydroclimatic regimes using basins

within the Western United States (USA).

To our knowledge, the systematic examination of model structure contributions to variations in flood frequencies is a novel contribution to the flood modelling literature. Previous work has examined uncertainty and sensitivities in statistical methods (e.g. Hosking and Wallis 1986; Stedinger et al 1993; Klemes 2000; Kidson and Richards 2005; Merz and Thieken 2005, 2009;

Hu et al. 2020), or from probabilistic hydrologic modelling systems (Hashimi et al. 2000; Franchini et al. 2000; Blazkova et





al. 2009; Arnaud et al. 2017). Specifically, the companion papers of Hashimi et al. (2000) and Franchini et al. (2000) undertake a one at a time local sensitivity analysis and a full sensitivity analysis using a factorial sampling design to examine basin climate characteristics and hydrologic model parameters impacts on FF estimates. Hashimi et al (2000) find that several parameters related to the basin climate (e.g. average rainfall, storm intermittency) along with several hydrologic model

parameters such as the percolation rate have higher sensitivity when considering FF estimates. They also conclude that soil moisture at event onset is the linking mechanism that explains why their particular parameters are the most sensitive. For example, soil moisture states closer to saturation result in larger floods for a given event with wetter soils modulated by a wetter mean climate or lower percolation rates. Franchini et al. (2000) perform a full sensitivity analysis and confirm the local sensitivity results. However, model structure is not systematically varied in Hashimi et al. (2000) and Franchini et al. (2000).


The overall goal of this study is to improve both the quality and efficiency of hydrologic risk estimates for infrastructure design. The specific objective is to understand which components of the modelling chain have the largest impact to FF estimates. To address this objective, we ask the following question: *What aspects of the modelling chain in stochastic FF analysis have the most sensitivity across a range of return intervals spanning 2-100,000 years*? Our null hypothesis is that for

rare floods (floods with return periods greater than 50,000 years) the sensitivity related to the precipitation event forcing dominates the total variance of a FF estimate as sketched in Figure 1a, with variance in FF estimates arising from the aforementioned factors: 1) model structure, 2) model parameters, 3) initial conditions, and 4) precipitation event forcing. We postulate that there may be other dominant factors contributing to FF sensitivity outside of precipitation event forcing for rare floods. We explore these components of the modelling chain by: 1) using a multi-hydrologic model ensemble, 2) sampling

model parameters across the model structures, 3) sampling model initial conditions that are internally consistent for each model structure from calibrated continuous long-term simulations, and 4) incorporating statistical uncertainty in the distributions that define the precipitation forcing. Further, we explore the impact of precipitation timing using two meteorological sequences; in one, we force the model with a single precipitation event, in the second, we force the model with a single precipitation event and random historical weather after the precipitation event to drive a stochastic (ensemble) event simulation framework. The

two different meteorological sequence methodologies were used to mimic different United States agency methodologies (Section 3.1.6). We use the analysis of variance (ANOVA) methodology to examine relative contributions of variance to FF estimates across the return periods of interest for all factors for both meteorological sequences. While the focus of this study was on stochastic rainfall-runoff modelling, the methods and implications discussed here may be applicable to simpler rainfall-runoff modelling as well, such as AEP-neutral model estimates.

**2. Study Basins**

The Island Park Dam in Idaho and Altus Dam in Oklahoma watersheds are used as representative basins of mountainous snowmelt (Island Park) and semiarid high plains (Altus) hydroclimates, respectively. These basins were selected because not



only are they representative of the dominant hydroclimates of the Western USA, they also have been the subject of past flood studies where basin delineations, observed streamflow, and precipitation frequency distributions were developed by

Reclamation.

Island Park (Figure 2a) is located on Henry's Fork River approximately 56 km north of Ashton, Idaho, and water stored at Island Park is used locally for irrigation. The Island Park watershed is roughly 1297 km$^2$ and includes steep mountain slopes along portions of the watershed boundary to nearly level slopes around Henrys Lake. Soils for the watershed range from low

permeability clays in the west to permeable volcanic sand in the east. There are areas within the watershed which are heavily forested and other areas which are barren. Elevations within the drainage area range from 1921 m at the crest of the spillway to 3231 m at Sheep Point along the northern boundary of the watershed (Reclamation 2015). Island Park has a strong seasonal cycle of precipitation, soil moisture, and streamflow with most of the watershed precipitation occurring as snow in October through May in the higher elevations. This results in a seasonal snowpack, maximized in late spring which then melts through

the summer, maximizing soil moisture and streamflow during late spring and early summer.

Altus Dam is on the North Fork Red River about 27 km north of the city of Altus, OK. The purposes of the dam and reservoir are to provide irrigation storage for lands in southwestern Oklahoma, flood control on the North Fork of the Red River, an augmented municipal water supply for the city of Altus, fish and wildlife conservation benefits, and recreation. The watershed

extends from Altus Dam in Oklahoma westward to Amarillo, Texas (Figure 2b). The watershed consists of generally rolling terrain with medium to coarse textured soils and spans an elevation from about 1120 m at the western edge of the basin to 450 m at the eastern outlet. This area contains many topographic features known as playa lakes (closed basins with a low area in the center that may see water storage following heavy rainfall) and thus the total contributing area is smaller than the total area of the watershed. We used the Reclamation estimated contributing area of 5051 km$^2$. Much of the basin above Altus Dam is

devoted to agriculture with a majority of the land cover consisting of cultivated crops, pasture, and hay production. The drainage basin contains no large forested areas, but there are treed riparian zones along the watercourses and trees in cultivated shelterbelts (Reclamation 2012). Altus Dam is a semi-arid basin that also has a seasonal cycle to precipitation with most occurring in winter through summer, primarily as rainfall. The spring and summer rainfall events are primarily convective in nature with sometimes very intense rainfall rates and high total accumulations over short periods of time that may coincide

with peak basin soil moisture in the spring.

## 3. Data and Methods

### 3.1 Modelling Workflow

Our stochastic hydrologic modelling workflow includes the Framework for Understanding Structural Errors (FUSE) hydrologic modelling framework, the Shuffled Complex Evolution (SCE) optimization algorithm, and precipitation frequency





distributions from Reclamation. Additionally, we have used the law of total probability (e.g. Tijms 2003; Nathan et al. 2003) and the analysis of variance (ANOVA) method to compute the FF estimates and partition the variance across the workflow components respectively.

For each basin, hydrologic models are configured and calibrated using an ensemble of historical meteorology (Newman et al.
2015). Then, long-term continuous simulations are made to generate spun-up initial conditions for event simulations. Event simulations are then performed across hydrologic models, model parameters, initial conditions, and precipitation frequency distribution estimates for two event sequence possibilities. For each precipitation frequency distribution, we split the probability density function into 50 bins and sample 25 events per bin and perform 2500 model simulations for each possible model-parameter-IC-precipitation frequency combination. This follows the total probability theorem methodology used at
Reclamation in their stochastic flood modelling. We implemented a factorial experimental design, using all combinations of the 10 hydrologic models, 11 parameter sets, 4 initial condition sets, and 11 precipitation frequency estimates for Island Park Dam (3 precipitation frequency estimates for Altus Dam) for a total 4840 combinations with 2500 model simulations per combination resulting in 12.1 million event simulations for Island Park Dam (referred to as Island Park) and 3.3 million event simulations for Altus Dam (referred to as Altus). The different precipitation frequency estimates come from the fact that this
project leveraged previously completed studies for these data. We do not believe this will significantly impact the results, as the ANOVA analysis takes these sampling differences into account.

## 3.2 Hydrologic Model Framework

The FUSE hydrologic modelling system is a freely available, modular modelling framework that enables developing and testing many conceptual hydrologic models in a single computational framework. It incorporates multiple parameterizations
for many hydrologic fluxes (or processes) at the individual flux level, with each equation formulated as a function of the model state, each in a separate code module. This allows the numerical solver to be separated from the flux parameterizations so that every FUSE configuration relies on the exact same numerical scheme. FUSE also incorporates a conceptual temperature index snow model, using elevation bands with user specified precipitation and temperature lapse rates to represent seasonal snowpack and changes in meteorology with elevation. Control at the individual flux level is key to understanding how changes in process
representation affect the modelling system behavior. See Clark et al. (2008) and Henn et al. (2015) for more details regarding FUSE.

FUSE uses several configuration files in which the user can specify the model decisions for process representation, numerical solver parameters, model calibration options, access to input and output data, etc. The structural modularity in FUSE is
underpinned by one file prescribing the equations to be used for each model component. This file can be changed independently from the other model settings, enabling the user to isolate the effects of the model structure decisions on the simulations. FUSE contains the SCE optimization algorithm (Duan et al. 1992) to calibrate any hydrologic structure the user specifies. SCE is a



robust global optimization algorithm that is widely used across the operational and research communities. FUSE uses the network common data format (netCDF) for all input and output data streams (forcing meteorology, any available observations for calibration, calibration results, simulated states and fluxes), with the same file formats regardless of hydrologic model configuration. Overall, the design of the FUSE system allows for easy configuration, calibration, and simulation of multiple hydrologic models for long term continuous simulations or short event simulations.

FUSE is first used to mimic three widely used hydrologic models: Hydrologic Engineering Center-Hydrologic Modelling System (HEC-HMS) model (Bennett 1998), the Variable Infiltration Capacity (VIC) model (Liang et al. 1994), and the SACramento-Soil Moisture Accounting (SAC-SMA) model (e.g. Anderson 2002). This provides a relatable base set of models to operational groups within the USA. Note that the FUSE instantiations of the models only mimic the actual models cited. FUSE does not use the same numerical solver, some process simplifications are made (particularly for VIC where we simplify evapotranspiration), different parameter estimations schemes are used, and FUSE does not contain the same coding errors as the original models (see Clark et al. 2008 for FUSE details). As a result, when mimicking a pre-existing model using a modular framework, some significant differences between their simulations can exist (Knoben et al., 2019). We then assembled seven other hydrologic model structures by varying particular processes from the three base models for a total of ten structures that we used to compute FF estimates for both basins (see Table 1 for the full list).

### 3.3 FUSE Meteorological Forcing and Calibration

All 10 hydrologic models for both basins were calibrated using the SCE optimization algorithm. We used KGE and RMSE as objective functions because the choice of objective function is subjective and dependent on available data and user needs. Additionally, recent work has highlighted that careful consideration needs to be given to the choice of objective function for high flow events (Mizukami et al. 2019). Root mean squared error (RMSE) is directly related to Nash-Sutcliffe Efficiency (NSE). Further, it can be shown that RMSE/NSE is made up of three component contributions to the total value: correlation ($r$), variability ($\alpha$), and bias ($\beta$). The Kling-Gupta Efficiency (KGE) is a reformulation of these same components, which allows the user to easily understand their individual contributions to the total KGE value (Gupta et al. 2009) and is shown in Equation 1.

$$ED_s = \sqrt{[s_r \cdot (r-1)]^2 + [s_\alpha \cdot (\alpha - 1)]^2 + [s_\beta \cdot (\beta - 1)]^2} \tag{1}$$

where $ED_s$ is the scaled Euclidian distance from the ideal point and $s_r$, $s_\alpha$, and $s_\beta$ are scale factors to adjust the weighting of the correlation, variability and bias terms (set to 1 typically). The KGE is also beneficial to use because the scale factors can be adjusted to emphasize the different components of KGE. Here we tested RMSE and KGE calibrations using daily streamflow, and KGE computed using annual peak flow values. We also examined modifying the KGE $s_\alpha$ scale factor from 1 to 5 to emphasize model flow variance in an effort to better capture flood peaks. Inflated $s_\alpha$ values resulted in model behavior





very similar to KGE using annual peak flows in agreement with Mizukami et al. (2019) and are not discussed further in Section
4.

A maximum of 40,000 model runs was allowed for the SCE calibration of each model structure and basin. Reconstructed daily
inflow data from Reclamation was used for Island Park, while annual peak flow data developed by Reclamation was used for
Altus due to lack of better available data for calibration at the time of this study. The impact of these different objective
functions and calibration data for the basins will be discussed in Section 4.

The meteorological forcing data consisted of a 100-member ensemble of gridded precipitation and temperature at 6 km
resolution described in Newman et al. (2015). Observations of precipitation and temperature and the process of projecting
point measurements to grids across sometimes complex terrain are inherently uncertain. This ensemble dataset was designed
to estimate those uncertainties and provide many plausible historical traces for hydrologic model applications. Each individual
member was used to calibrate each hydrologic model, resulting in a 100-member ensemble of calibrated model parameters for
each model for each basin (100 ensembles × 10 models × 2 basins). Because of the available observational data, different spin
up and calibration periods were used. For Island Park, the hydrologic models were spun up for water years (WY) 1970-1979
and calibrated on WY 1980-2014 (35 WYs), while Altus was spun up for WY 1980-1984 and calibrated on WY 1985-2011
(27 WYs). Again, while the number of WYs for both catchments is similar, data availability meant that Altus calibration only
relied on annual peaks, while for Island Park daily streamflow values were used.

### 3.4 Initial Condition Specification

Continuous simulations using the subsampled parameter sets were then performed and full model states were output each day
for the full calibration periods for each hydrologic model and basin. These states were sampled to determine the ICs for the
event simulations. Sampling initial states from continuous simulations has the advantage of providing ICs that are consistent
with the specific hydrologic model and parameter set. Applying random perturbations to an IC may result in unrealistic states
and subsequent simulation results.

For Island Park, the focus was on ICs from April through June that had minimal (> 10 mm) snow water equivalent snowpack
to represent flood events near the end of the snowmelt season around peak climatological soil moisture storage. For Altus, the
focus was on late winter through mid-summer ICs (February-July) when both soil moisture and precipitation event intensity
and volumes are around their climatological maximums. For both basins and all models, four ICs were sampled in the top 10
percent, the 90th, 94th, 97th, and 99th percentiles of total column soil moisture within all validation years and months.

### 3.5 Precipitation Frequency Estimates

Regional frequency analysis (RFA) is a useful method for extending the period of record in environmental datasets by means
of a "space-for-time" substitution where additional information in space supplements the lack of information in time. The





basic assumption of RFA is that extreme events recorded at stations located within a predetermined homogeneous region can be described by the same probability distribution. By scaling the data by the respective at-site mean (ASM), the user assumes that a single probability distribution is valid for every location within the homogeneous region, while the magnitude can vary spatially.


The L-moments method (Hosking and Wallis 1997) is one example of a regional frequency method. The basis of the L-moments algorithm is that linear combinations of moments can be used to estimate model parameters for extreme value distributions. The moments of interest (also referred to as L-statistics) include L-CV, L-skewness, and L-kurtosis and are computed for every site utilized in an analysis. Regional moments are developed using weighted averages of the site-specific moments, where the weight is proportional to period of record. The regional L-moments are then used to define the regional growth curve (RGC), a dimensionless quantile function that represents the cumulative distribution function of the frequency distribution valid for all sites located within the homogenous region. Site-specific precipitation-frequency estimates ($Q_i(F)$; Equation 2) are developed by scaling the RGC ($q(F)$) by a site-specific ASM ($\mu_i$), allowing the magnitudes of precipitation-frequency estimates to vary spatially across the region of interest.


$$Q_i(F) = \mu_i q(F), \qquad\qquad\qquad\qquad\qquad\qquad (2)$$

Reclamation (2015) developed median and uncertainty precipitation-frequency curves for the Island Park watershed using a regional L-moments approach combined with Latin hypercube resampling procedures. More specifically, the authors used annual maximum two-day precipitation totals from 45 stations in a homogeneous region surrounding the Island Park watershed to estimate parameters of the four-parameter Kappa distribution. The authors used Latin-hypercube sampling methods in R via the "qnorm" function to perform Monte Carlo sampling to create 300 parameter sets using variations in five parameters: at-site mean, regional L-Cv, regional L-skew, regional L-kurtosis, and areal-reduction factor. Results from this analysis include 11 frequency distributions 5th, 14th, 23rd, 32nd, 41st, 50th, 59th, 68th, 77th, 85th, and 95th percentiles. Kappa parameters from Reclamation (2015) are reproduced in Table 2. During stochastic simulations performed here, we force two-day historical precipitation events to equal basin-average magnitudes sampled from the two-day precipitation frequency curve valid over the Island Park watershed, while retaining the spatial precipitation structure from the historical event.

Similarly, Reclamation (2012) developed precipitation-frequency estimates including median and uncertainty bounds for the Altus watershed using a regional L-moments approach combined with Latin hypercube sampling procedures. The authors focused on annual maximum one-day (or 24-hour) precipitation totals recorded at 482 stations with at least five years of data and used Latin hypercube sampling to produce 150 parameter sets based on variations in the same five parameters listed above: at-site mean, regional L-Cv, regional L-skewness, regional L-kurtosis, and areal-reduction factor. The report provides all precipitation-frequency estimates in the form of fourth-order polynomials, with coefficients reproduced in Table 3. Similar to





Island Park simulations, we force one-day historical precipitation events to equal basin-average magnitudes sampled from the one-day precipitation frequency curve valid over the Altus watershed, while retaining the spatial precipitation structure from the historical event.

### 3.6 Event Sequencing

Some stochastic modelling studies at Reclamation force the rainfall-runoff model with a precipitation event (e.g., two-day

event) followed by no precipitation for the remaining simulation time (Reclamation 2018). The lack of additional precipitation after the primary precipitation event is not based on any physical reasoning, thus we examine both dry and historical meteorological sequences after the primary precipitation event (two-day at Island Park and one-day at Altus). The forcing event lengths differ because of the differing meteorology driving floods in the two basins. Again, more intense shorter duration convective precipitation events primarily cause flooding at Altus, while longer duration precipitation events associated with

extratropical cyclones is the primary flood driver at Island Park. In the dry meteorological sequence, we set precipitation to zero after the primary precipitation event. In the historical meteorology setup, we randomly sample ensemble member meteorology sequences using the same start date corresponding to the sampled IC for the simulation period only. In other words the ICs are taken from a continuous simulation and define the event start date, but we then randomly sample from all 100 members to redefine the event period. In both cases, the primary precipitation forcing (two-day and one-day) is forced to

equal sampled values from the precipitation frequency curve. Future work should examine event sequencing in greater detail, particularly to quantify the impacts of possible future circulation changes on FF estimates and sensitivities.

### 3.7 ANOVA

As noted above, the total probability theorem is used to compute modelled basin runoff at return periods of 2, 5, 10, 20, 50 100, 500, 1,000, 5,000, 10,000, 50,000, and 100,000 years from the stochastic simulations for all model, parameter, IC, and

precipitation distribution combinations, for both event sequences. An ANOVA analysis is then performed on the runoff values for all the return periods for both event sequences and basins. The ANOVA framework is a computationally frugal way to estimate individual component contributions to the total variance of a variable such as runoff by relying on a sum of squares decomposition. ANOVA analyses have been widely used in hydrometerology to separate the components of future climate changes (Hawkins and Sutton, 2009; Lehner et al., 2020) and to determine which elements of the model chain contribute most

to the spread of the projected changes in streamflow (Bosshard et al., 2013; Addor et al. 2014; Breuer et al., 2017; Chegwidden et al., 2019).

By estimating the fractional (relative) variance contributions of each factor and all two factor interactions, we identified the pieces of the modelling workflow which contribute most to FF sensitivity for each return period. We used the 'anovan'

MATLAB function as implemented in MATLAB version 9.8.0.1380330 (2020a) Update 2. This function allows for N-way ANOVA computations with mixed continuous and categorical predictors, and specification of the interaction terms to be





estimated (https://www.mathworks.com/help/stats/anovan.html). Here we specify model structure and parameters as categorial predictors and precipitation event forcing and initial conditions as continuous predictors. Precipitation event forcing and initial condition values are normalized before the ANOVA analysis was performed. Finally, we perform a subsampling

and bootstrapping of the effects that have more samples than the effect with the fewest samples (e.g. for Island Park ICs have 4 samples, precipitation frequency distributions have 11 samples) following Bosshard et al. (2013). Disparate sample sizes can bias the fractional variance estimates, overestimating the contributed variance for effects with more samples. Performing subsampling with bootstrapping (n=1000) alleviates the bias (Bosshard et al. 2013).

## 4. Model Calibration

When examining daily flow time series, the KGE and RMSE daily metrics produce more realistic simulations than the KGE interval metric as seen in Figure 3. This is a somewhat expected result as the interval metric contains no time information (correlation) on the daily scale. The daily KGE metric based calibration outperforms the daily RMSE based calibration, where the daily RMSE based calibration underestimates the flow variance (not shown) in agreement with past studies (Gupta et al. 2009). The KGE interval metric-based calibration represents the peak flows well (with some overrepresentation) but has large

differences in event recession curves with overestimation of flow in the days and weeks immediately following high flow events. This erroneous recession curve representation would result in very different volume-based floods versus daily metric-based calibrations.

Given the above calibration characteristics and the available calibration data at Island Park (daily flow) and Altus (annual peak

flow), daily KGE was selected as the calibration metric for Island Park and interval KGE as the calibration metric for Altus. Daily KGE provides the best all-around simulation when considering daily peak flows as well as volume integrations over days to weeks at Island Park. For Altus, calibrating to yearly peak flows using KGE provided a better overall peak flow calibration than RMSE calculated using annual peak flows, likely due to the reformulated weighting of bias and variance as compared to RMSE. Again, these results agree with Mizukami et al. (2019), which examined some of the same calibration

metrics using multiple hydrologic models and hundreds of basins across the contiguous United States. They found that KGE outperforms RMSE (or NSE) based calibrations and that peak flow metrics do outperform KGE for peak flow simulation but result in much degraded daily model performance with sometimes severe modelled flow biases.

Figure 4 highlights the final CDF of the calibrated KGE for all ten models for Island Park (Fig. 4a) and Altus (Fig. 4b). It is

not possible to make direct performance comparisons between the models at the two basins given that the KGE values are based on daily (Island Park) and annual peak runoff (Altus). However, in a broad sense, model behavior at Island Park is much more constrained than Altus based on the relative ranges of calibration scores for each basin (different x-axis ranges from left





to right panels). These differences informed the model parameter sampling strategies and show that the model behavior at Island Park is more constrained than at Altus along the model parameter dimension.

## 4.1 FUSE Parameter Set Selection

The 100 parameter sets available for each model and basin were subsampled for the final FF event simulations. Because Island Park had more available data for calibration, the final calibrated model performance was very similar across the 100 members for all 10 hydrologic models. Therefore, 11 parameter sets spanning the full range of model performance were sampled for each hydrologic model using the 1st, 10th, 20th, 30th, 40th, 50th, 60th, 70th 80th, 90th and 99th percentiles of the cumulative density function (CDF) of the calibration objective function. For Altus, the calibrated model behavior was less constrained due to the much smaller amount of calibration data available. Therefore, the 10 best calibrated parameter sets for each hydrologic model were used, which constrained model parameter induced differences in model behavior, but still not to the same level as Island Park.

## 5. Sensitivity Analysis

The ANOVA analysis was performed using the full complement of FF estimates for both basins and precipitation event forcing sequences. All fractional variance contributions are normalized by the total variance in the FF estimate for each return period such that if a component has a fractional variance of 0.5 that component contributes half of the total variance for that return period. The plots represent the 2, 5, 10, 50, 100, 1,000, 10,000, 50,000, and 100,000-year return periods. For Figures 6 through 11, the dry meteorological sequence is always in panel a) and the historical meteorology event sequence is always in panel b), and the color coding follows Figure 1. Interaction terms are a blend of the two primary components (e.g. model structure-model parameter interactions are red-orange).

Normalized FF plots including all possible effect combinations for both models are shown in Figure 5. Annual exceedances at Island Park in the mean follow a nearly linear trend on the semi-log X-axis plot with the range of possible values having relatively higher spread at larger return intervals (Fig. 5a), which is consistent with the hydrology of Island Park being a less flashy more snowmelt flow dominated basin. The normalized FF curve at Altus is highly non-linear even with a semi log X-axis with little flow for many small return periods (Fig. 5b). Sharp increases in flood responses after roughly the 500 year return period are seen with normalized spread larger than at Island Park for the largest return periods (50-100,000 years).

### 5.1 Island Park

ANOVA results using all available model structures, sampled parameter sets, sampled ICs, and sampled precipitation frequency distributions for Island Park are shown in Figure 6. When all model structures are included ICs dominate the frequent events less than about 5000 years, while the precipitation frequency distribution is the most important for rarer events.





Model structure consistently contributes roughly 20% of the variance and is generally the second most important effect across all return periods, outside of 1,000 – 10,000 year flood where ICs, precipitation frequency curves, and model structure vary in leading, secondary or tertiary importance depending on the dry or wet meteorological sequencing. Model parameters and interaction terms contribute roughly 10% of the variance for less frequent events for both meteorological sequences. For rare floods with return periods larger than 50,000 years, event precipitation is about twice as important as model structure and 3 times more important than ICs for dry sequences after the event input, while for historical meteorological sequencing, the event forcing is only about 1.5x more important than model structure for 100,000 year floods.

Figure 7 presents the fractional variance contributions for Island Park using the three base models: HEC-HMS, VIC, and SAC-SMA. Similar to all models, ICs and the precipitation frequency distribution specification are the most important for frequent and extreme events, respectively. Model structure is the second most important contributor for frequent events, but for return periods larger than 1,000 years, model structure-parameter interactions become as or more important than model. Again, moving from the dry to historical meteorological sequence decreases the variance contribution of precipitation frequency distributions, and increases the importance of model structure, model structure-parameter interactions, and ICs across all return periods (compare Figure 7a to 7b). This is somewhat counter intuitive but may be related to the fact that soil states can strongly influence recession curve characteristics and additional non-extreme precipitation event forcing is either stored or released within the 14 day volume integration depending on model structure, parameters, or ICs.

Using a different combination of the ten possible model structures results in a slightly different conclusion. The set of simulations presented in Figure 8 represents the set of three hydrologic models that generates the largest flood responses to larger precipitation event forcing. Overall, the precipitation frequency distribution specification is still the most important at extreme events, and ICs are most important for very frequent events, but model structure contributes a larger fraction of the total variance across all return periods and is often of similar magnitude to either ICs or precipitation frequency distribution changes (Figure 8). Here we see that moving from the dry to historical meteorological sequence increases the importance of model structure (compare Figure 8a to 8b). This is because these three model structures have more variation between each other given additional precipitation input than the variability in runoff changes due to ICs. Differences in surface runoff versus subsurface storage and slower baseflow appear to be driving the model structure variability and is discussed more in Section 6.

## 5.2 Altus

The ANOVA results using all available model structures, sampled parameter sets, sampled ICs, and sampled event forcings for Altus are shown in Figure 9. Similarly to Island Park, ICs are most important for frequent events, while precipitation event forcing is most important for rarer events. Two differences are of note here. First, precipitation event forcing is generally





more important across return periods at Altus versus Island Park. Second while model structure is slightly less important, model parameters and model structure-parameter interactions are of similar importance to model structure, such that the combination of model structure and parameter effects and interactions is as important as precipitation event forcing for both meteorological sequences. Finally it is evident that meteorological sequencing is inconsequential at Altus, which makes
intuitive sense given the single day peak flow metric for Altus versus the 14 day integrated volume metric at Island Park.

The ANOVA results for Altus using the three base models show a similar picture as for Island Park. ICs almost always contribute the most variance for frequent events (less than a few hundred years) and the precipitation frequency distributions are the most important for larger events (Figure 10). However, the precipitation frequency distributions are even more
important for Altus than at Island Park particularly for the historical meteorological sequence, as they contribute around 50% of the total variance for 50,000-100,000 year events as compared to around 30% at Island Park. Model structure and model structure-model parameter interactions are of secondary importance across essentially all return periods. Again, moving from dry to historical meteorological sequencing does not change the picture significantly at Altus (compare Fig 10a to 10b), which is expected as the flood metric is the single day maximum flow and generally single day maximum flow is directly related to
the extreme precipitation flood event input and not subsequent smaller events.

Further examination of multiple model combinations at Altus revealed that using the two most disparate model responses, SAC-SMA (Model #3) and the SAC-SMA/HEC-HMS combination (Model #6) models results in substantial increase in importance of model parameters and model structure – model parameter interactions (Figure 11). In fact, model structure –
model parameter interactions contribute the most variance across all return periods in this case. Additionally, model structure and model parameter effects contribute similar variance to the precipitation frequency distributions. Again, moving from dry to historical meteorological sequencing does not substantially change the message here as expected (compare Figure 11a to 11b). For this case the model responses are starkly different, such that it may be possible to rule out one of the model structures as plausible, however model structure selection work is outside the scope of this study.

**6. Discussion**

The results of this study demonstrate that workflow and methodological decisions impact hydrologic model behavior and the final variance estimates of a FF study. This suggests that careful consideration of the various components of stochastic flood modelling should be undertaken. To our knowledge, the inclusion of model structure into FF estimate sensitivity analysis is a novel contribution to our understanding of stochastic flood modelling systems. We reaffirm that calibration metrics only
constrain model behavior for components of the hydrograph most related to the calibration metric (e.g. Mendoza et al. 2015, Mizukami et al. 2019). For streamflow-based calibration, KGE is a robust metric that provides balanced model behavior across all components of the hydrograph because of its formulation and should be used over RMSE/NSE if possible.



Furthermore, calibration metrics focusing on high flow only generally result in degraded model performance for other parts of the hydrograph such as the recession curve, in agreement with Mizukami et al. (2019). The implication for this work is that
calibrated hydrologic models using RMSE/NSE may have inferior performance for longer duration volume flood metrics because of substantial biases introduced during calibration that was not designed to constrain flow volumes.

The ANOVA results demonstrate that ICs contribute the most variance for frequent events and the precipitation frequency distribution specification contributes the most variance for extreme events. One area for future study is the specification of
the precipitation frequency distribution and uncertainty estimates. Here we relied on previously published precipitation frequency results, as developing new estimates is outside the scope of this study. However, it is possible that the specification of the distribution and the uncertainty estimation methodology could have an impact on subsequent analysis steps. Furthermore, the precipitation frequency distribution methods differ across the basins, which is an inconsistency with unquantified impacts. Normalization of the precipitation inputs before the ANOVA analysis possibly mitigates these potential
issues, but further exploration could be undertaken in future work.

Additionally, model structure, model parameters, and model structure-parameter interactions may have important contributions across the return periods depending on the flood metric and basin. In this study all ten model structures are treated as equally plausible. Future stochastic FF studies should consider model structure in their experimental design with thought given to
constraining the model structure ensemble to plausible model configurations using available techniques (Jakeman and Hornberger 1993; Gupta et al. 2012). Model parameter and model structure – model parameter interactions are more important at Altus, where the available calibration data limited the ability for calibration to constrain model performance. Consideration of model parameter variations should be taken into account when scoping projects with little calibration data available.

Differences in model total storage and subsequent runoff generation drive the different flood responses across both basins. Figure 12 shows the average model response for models #1 (HEC-HMS) and #3 (SAC-SMA) for a subset of precipitation event forcings for Island Park. The change in storage and cumulative runoff are normalized by the total precipitation event forcing to highlight storage and runoff efficiency differences between the two models. Note the precipitation event forcing occurs on days 1 and 2. Models with high event-based runoff ratios generate runoff more readily and have smaller subsurface
storages, while models with lower event runoff ratios allow for more infiltration and storage. Model #3 stores about 60% more of the precipitation event than model #1 and ends up generating 25% less cumulative runoff than model #1. These differences are more important for basins using integrated flood metric such as Island Park here, as responsive models generate larger volumes while the other models store more of the precipitation event forcing and release it over longer periods of time. This point should be the focus of additional study and provides one physical process comparison to identify the appropriate model
structures for a given basin.





While the focus of this study was on stochastic rainfall-runoff modelling for FF studies, there are potentially broader applications to hydrologic modelling for any purpose, including planning, design, or restoration often focused on more frequent floods up to extreme events for risk analysis. For example, stochastic rainfall-runoff modelling is data and labor intensive, thus less intensive methods are frequently used, most commonly AEP-neutral assumptions of precipitation return period being equal to flood return period. Even in those studies, model selection, parameterization, initial conditions, calibration, and forcing still play an important role in model outcome. Additionally, examining a range of return periods rather than just extreme floods was intentional to help inform a broader range of applications beyond those focused on risk for large dams where only extreme events are relevant. Understanding of sensitivity in rainfall-runoff modelling, whether stochastic or not, is important for flood studies. The results of this study can help guide model selection and development and provide a better understanding of variance in a variety of flood studies.

## 7 Conclusions

The key generalizable conclusions are:

1) ICs and precipitation frequency distributions contribute the most variance in the stochastic flood modelling chain for frequent and extreme events respectively.

2) Hydrological model structure can be equally important, particularly for multi-day volume flood metrics. This highlights the need to critically assess assumptions underpinning models, understand basin flood generation processes and develop methods to select appropriate models. This includes the re-examination of the AEP neutral assumption and shifting to model process parameterizations that are most plausible for the study catchment.

3) Model parameter and model structure-parameter interactions can be important if the model parameter space is not well constrained during calibration.

4) Confirming many other studies (e.g. Gupta et al. 2009, Mizukami et al. 2019), the Kling-Gupta Efficiency (KGE) results in better hydrologic model performance than NSE (or RMSE) for calibration of extreme events and volume integrated flood metrics, and is more flexible for application specific uses through the use of user specified component weights.

## Data Availability

The watershed characteristics, reconstructed flow data, and precipitation frequency estimates are available by request from the United States Bureau of Reclamation. The FUSE model output data are available at https://data.usbr.gov.

## Code Availability

The FUSE model is available at https://github.com/anewman89/fuse.



**Author Contribution**

All authors contributed to the experimental design. AJN and MS performed the computations while AJN, AGS, and KDH prepared the figures and tables. AJN led preparation of the manuscript with contributions from all authors.

**Competing Interests**

The authors declare that they have no conflict of interest.

**Acknowledgements**

The U.S. Bureau of Reclamation Science and Technology program funded this work under award number R16AC00039, and it was supported by the National Center for Atmospheric Research, which is a major facility sponsored by the National Science Foundation under Cooperative Agreement No. 1852977. We gratefully acknowledge high-performance computing support

from Cheyenne (http://10.5065/D6RX99HX) provided by NCAR's Computational and Information Systems Laboratory, sponsored by the National Science Foundation.

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




**Table 1. FUSE hydrologic processes (far left column) and the various selected process representations for the ten hydrologic models.**

| FUSE Config. | HECHMS | VIC | SACSMA | MODEL4 | MODEL5 | MODEL6 | MODEL7 | MODEL8 | MODEL9 | MODEL10 |
|---|---|---|---|---|---|---|---|---|---|---|
| **rainfall error** | multiplc_e | multiplc_e | multiplc_e | multiplc_e | multiplc_e | multiplc_e | multiplc_e | multiplc_e | multiplc_e | multiplc_e |
| **upper-layer architecture** | tension1_1 | onestate_1 | tension1_1 | tension2_1 | onestate_1 | tension2_1 | onestate_1 | tension1_1 | onestate_1 | tension1_1 |
| **lower-layer architecture and baseflow** | unlimfrc_2 | fixedsiz_2 | tens2pll_2 | unlimfrc_2 | unlimfrc_2 | unlimpow_2 | tens2pll_2 | tens2pll_2 | tens2pll_2 | unlimfrc_2 |
| **surface runoff** | arno_x_vic | arno_x_vic | prms_varnt | arno_x_vic | arno_x_vic | prms_varnt | prms_varnt | prms_varnt | prms_varnt | arno_x_vic |
| **percolation** | perc_f2sat | perc_w2sat | perc_lower | perc_f2sat | perc_f2sat | perc_lower | perc_lower | perc_f2sat | perc_w2sat | perc_lower |
| **evaporation** | sequential | rootweight | sequential | sequential | sequential | sequential | sequential | sequential | rootweight | sequential |
| **interflow** | intflwnone | intflwnone | intflwsome | intflwnone | intflwnone | intflwsome | intflwsome | intflwnone | intflwnone | intflwsome |
| **time delay in runoff** | rout_gamma | rout_gamma | rout_gamma | rout_gamma | rout_gamma | rout_gamma | rout_gamma | rout_gamma | rout_gamma | rout_gamma |
| **snow model** | temp_index | temp_index | temp_index | temp_index | temp_index | temp_index | temp_index | temp_index | temp_index | temp_index |






**Table 2. Parameters used to define the four-parameter Kappa distribution. Table reproduced from Table 4.5 in Reclamation**
**(2015).**

| Sim | Percentile | xi | alpha | K | H | Basin Mean |
|---|---|---|---|---|---|---|
| 1 | 95th | 0.8059 | 0.02842 | -0.068 | 0.1374 | 1.66 |
| 2 | 85th | 0.8083 | 0.2827 | -0.0635 | 0.1235 | 1.64 |
| 3 | 77th | 0.8108 | 0.2812 | -0.0590 | 0.1095 | 1.63 |
| 4 | 68th | 0.8132 | 0.2798 | -0.0546 | 0.0956 | 1.61 |
| 5 | 59th | 0.8157 | 0.2783 | -0.0501 | 0.0816 | 1.6 |
| 6 | 50th | 0.818 | 0.2768 | -0.0456 | 0.0676 | 1.58 |
| 7 | 41st | 0.8188 | 0.2768 | -0.0395 | 0.0634 | 1.57 |
| 8 | 32nd | 0.8195 | 0.2768 | -0.0334 | 0.0592 | 1.55 |
| 9 | 23rd | 0.8203 | 0.2767 | -0.0272 | 0.0549 | 1.54 |
| 10 | 14th | 0.821 | 0.2767 | -0.0211 | 0.0507 | 1.52 |
| 11 | 5th | 0.8217 | 0.2767 | -0.0430 | 0.0463 | 1.51 |

**Table 3. Polynomial coefficients (fourth-order) that describe the lower, median, and upper precipitation-frequency curves for Altus. Table reproduced from Table 5.7 in Reclamation (2012).**

| | A0 | A1 | A2 | A3 | A4 |
|---|---|---|---|---|---|
| Lower Estimate (5%) | 0.906821 | 0.359010 | 0.031004 | 0.009728 | -0.000563 |
| Median Estimate (50%) | 0.999012 | 0.391658 | 0.033909 | 0.013662 | -0.000692 |
| Upper Estimate (95%) | 1.082307 | 0.426903 | 0.04651 | 0.017021 | -0.000828 |






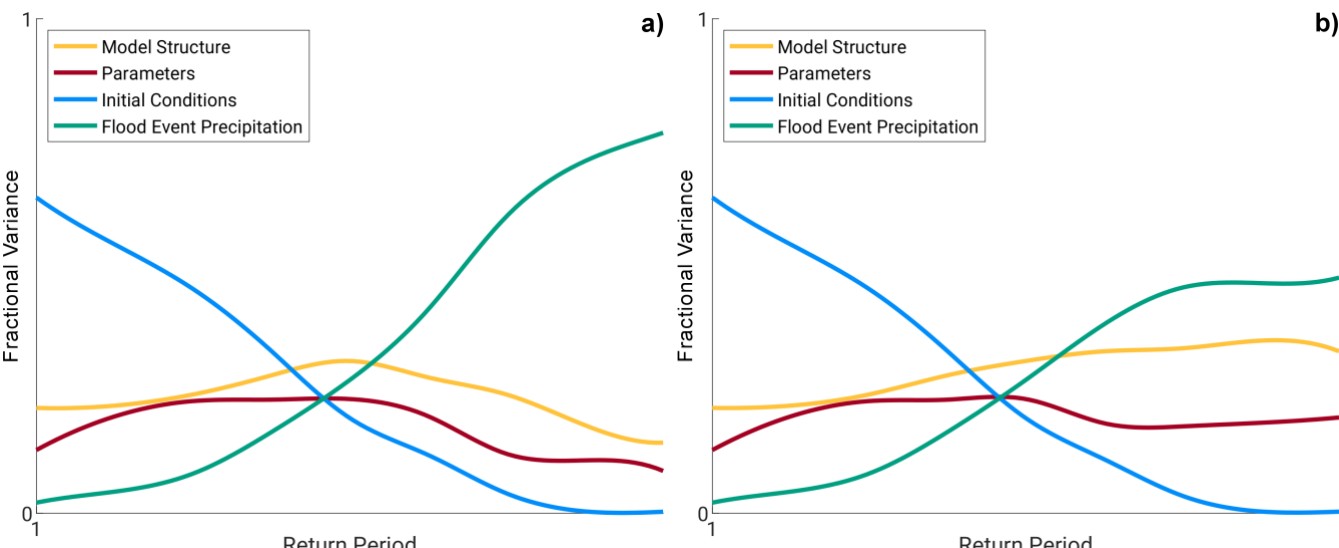

**Figure 1. Conceptual contribution of relative variance contribution from initial conditions (blue), model parameters (red), model structure (orange), and precipitation event forcing (green) across return periods (larger return periods towards right) for a) the base case and b) one possible alternative where model structure has similar importance to precipitation event forcing for extreme events.**

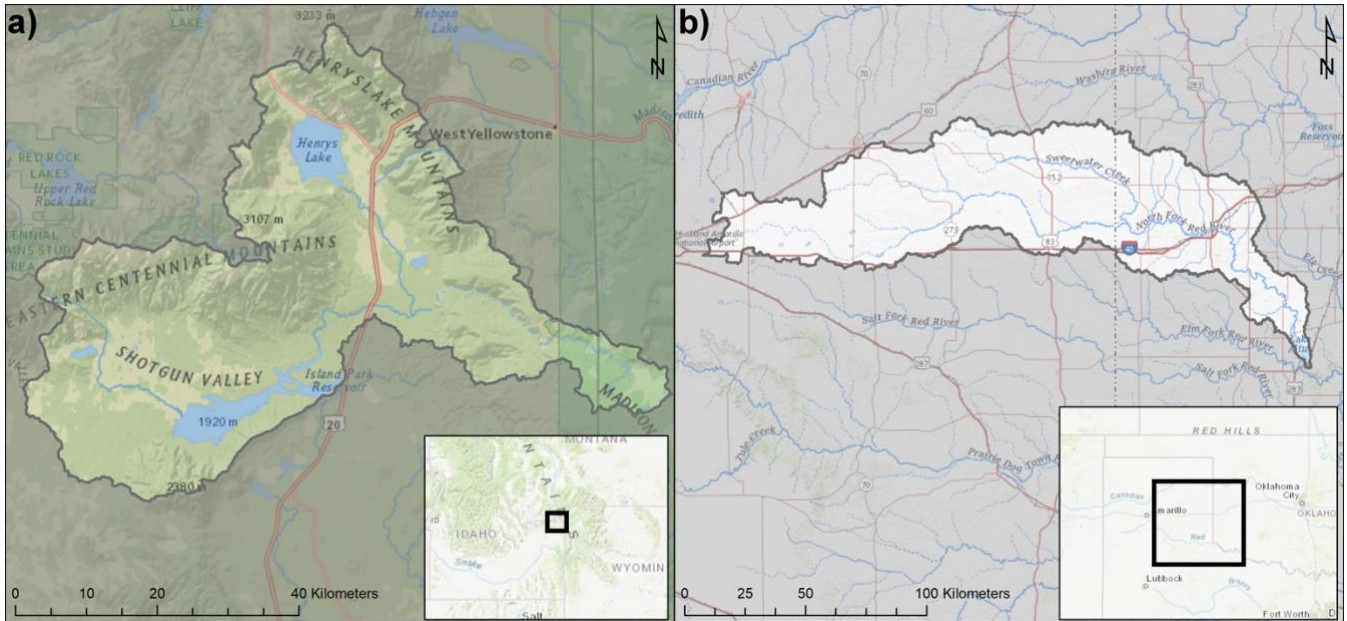

**Figure 2. a) Island Park and b) Altus watershed locations. Base layers © esri (Environmental Systems Research Institute)**







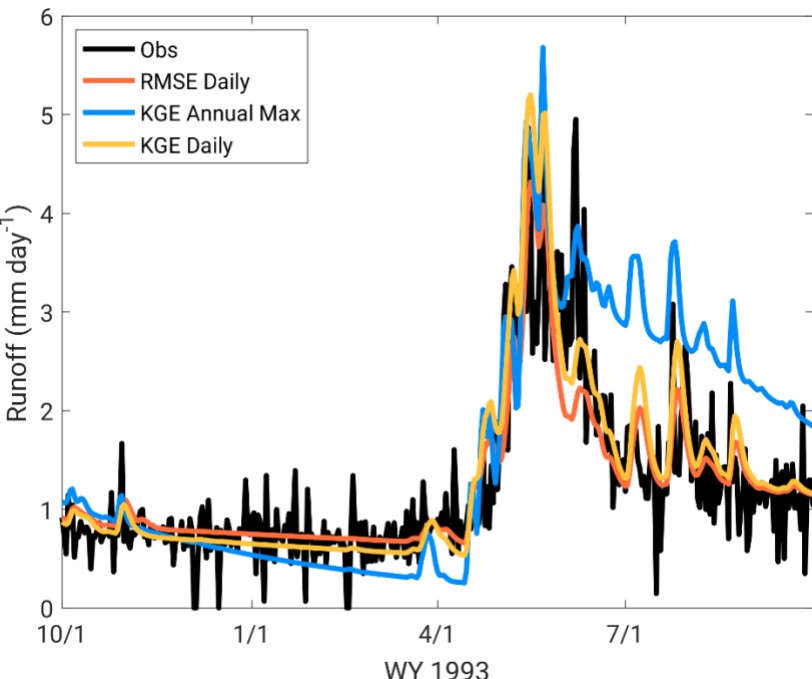

**Figure 3. Island Park calibration period runoff for water year (WY) 1993 with RMSE using daily flow, KGE using daily flow, and KGE using annual maximum flow.**

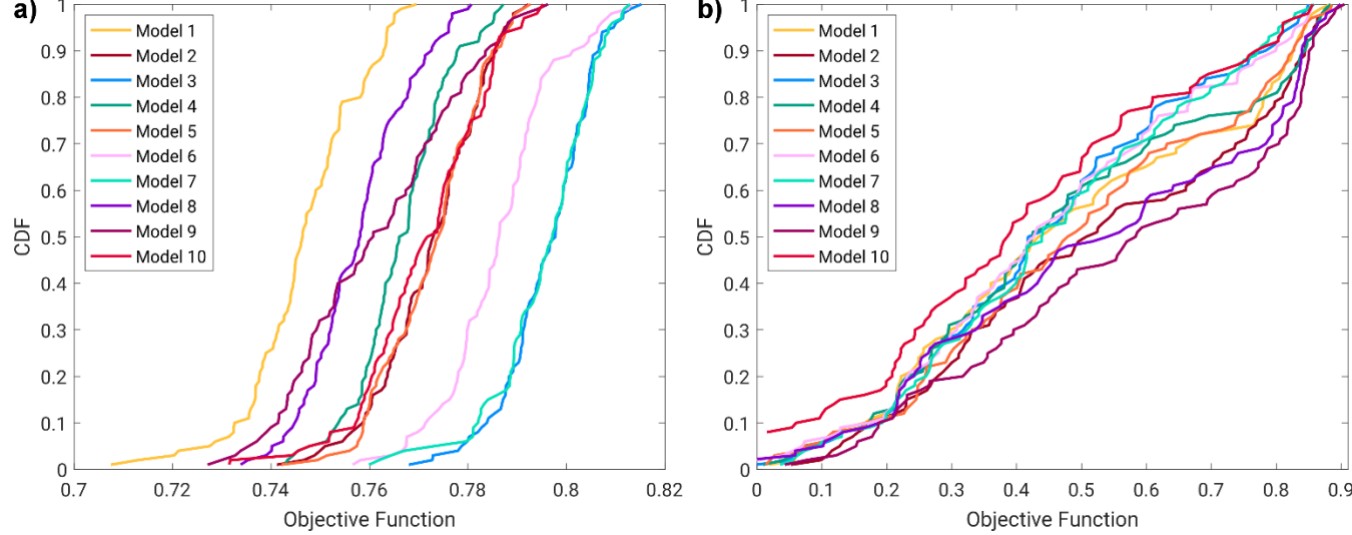

**Figure 4. a) Island Park daily flow calibrated KGE distributions for all 10 models and b) Altus yearly peak flow calibrated KGE distribution for all ten models.**


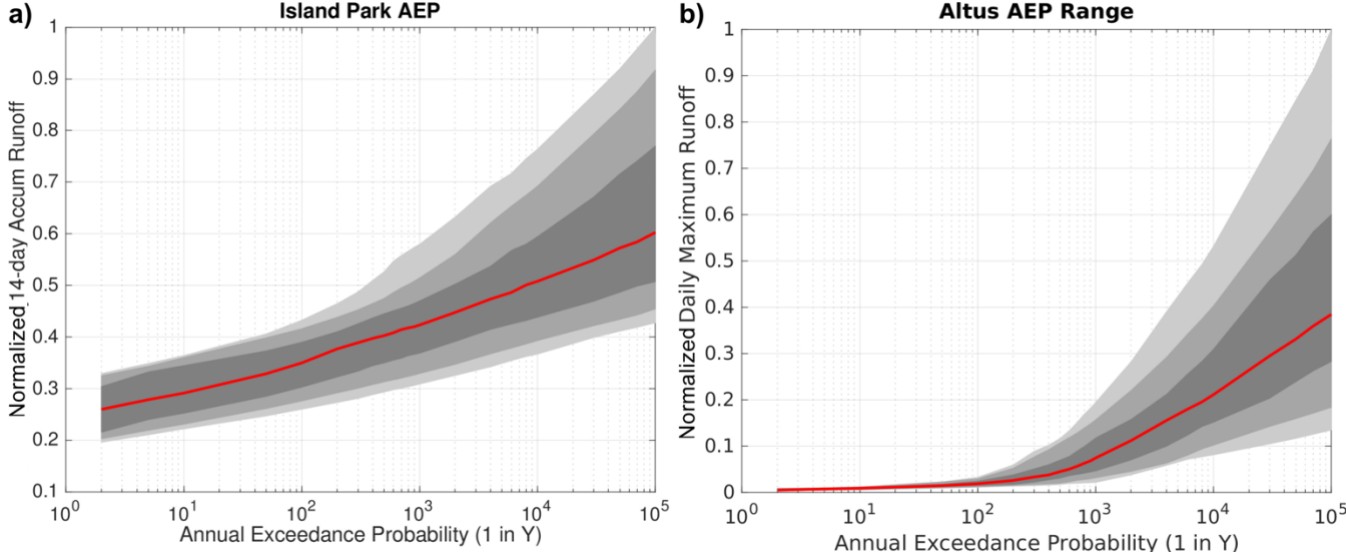

**Figure 5.** Normalized (by maximum possible flood runoff) FF curves with the median in red, and the interquartile range (25th-75th percentiles) in dark gray, 10th-90th percentile spread in medium gray, and the minimum to maximum spread in light gray for a) Island Park and b) Altus.

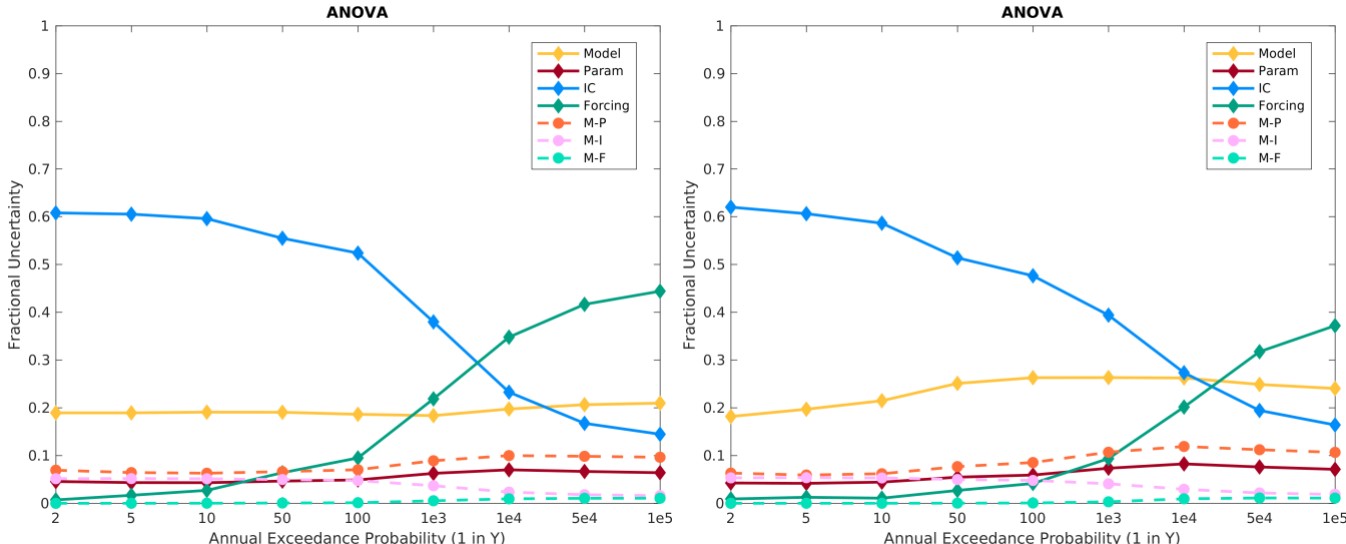

**Figure 6.** Island Park fractional variance contributions using all ten model structures for the a) dry meteorological sequence and b) historical meteorological sequence. Only interaction terms that contribute significant variance are shown in Figures 6 through 11.





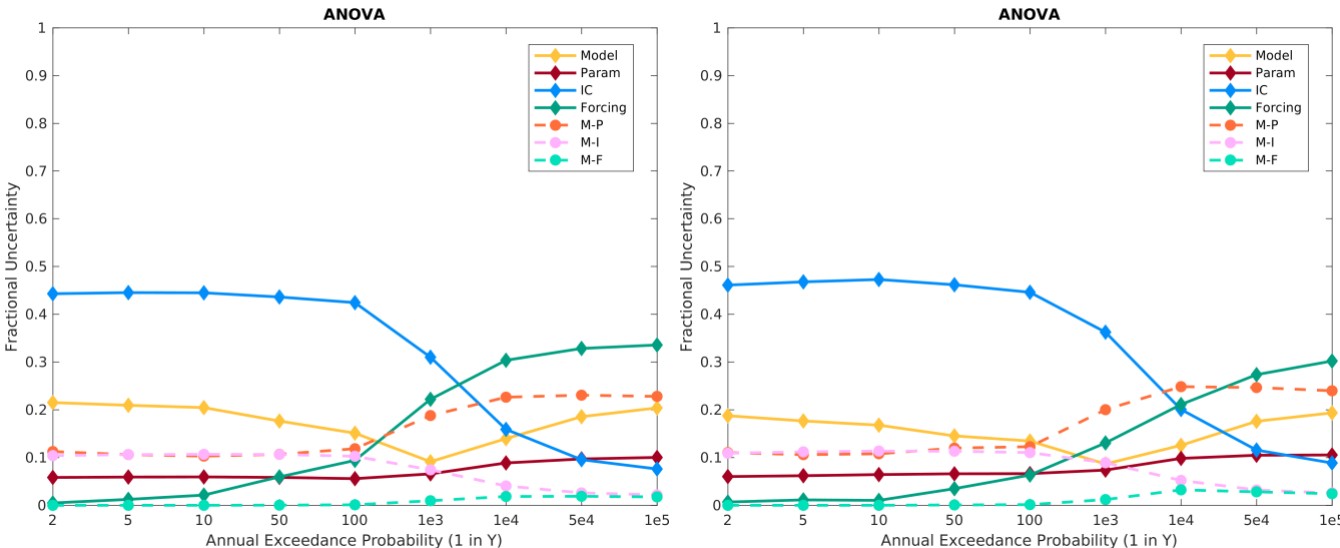

**Figure 7. Island Park fractional variance contributions using the three base models: HEC-HMS (Model #1), VIC (Model #2), SAC-SMA (Model #3), for the a) dry meteorological sequence and b) historical meteorological sequence.**

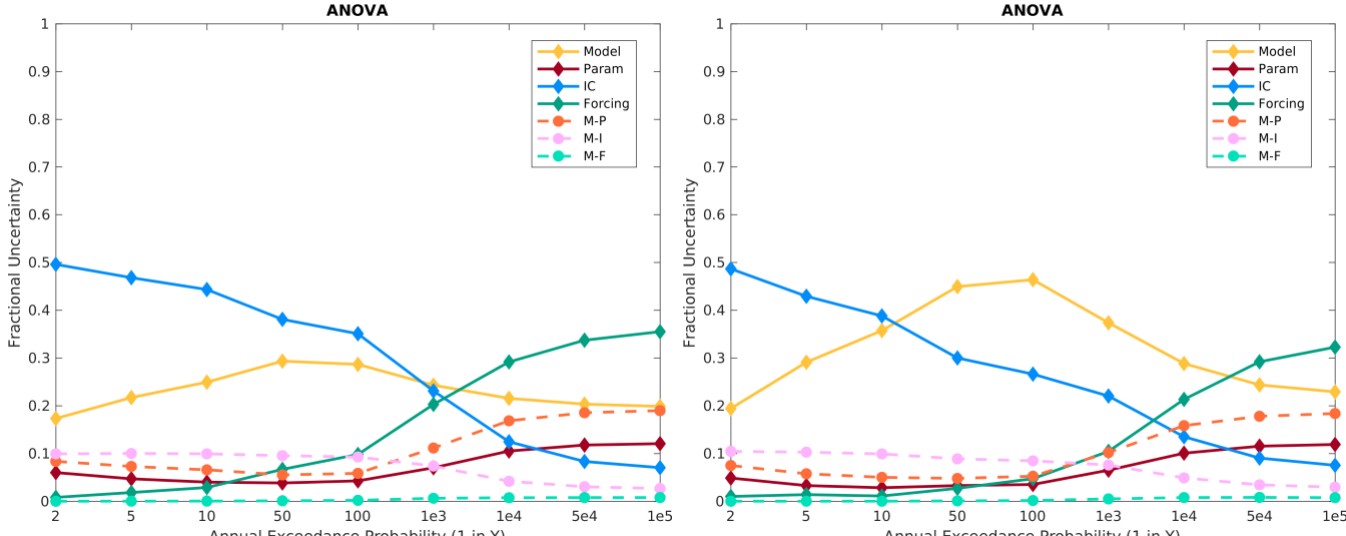

**Figure 8. Island Park fractional variance contributions for the three most responsive model structures (i.e. structures associated with the largest runoff/precipitation ratio): HEC-HMS (Model #1), HEC variant (Model #4), and a SAC-SMA/HEC-HMS combination (Model #6), for the a) dry meteorological sequence and b) historical meteorological sequence.**



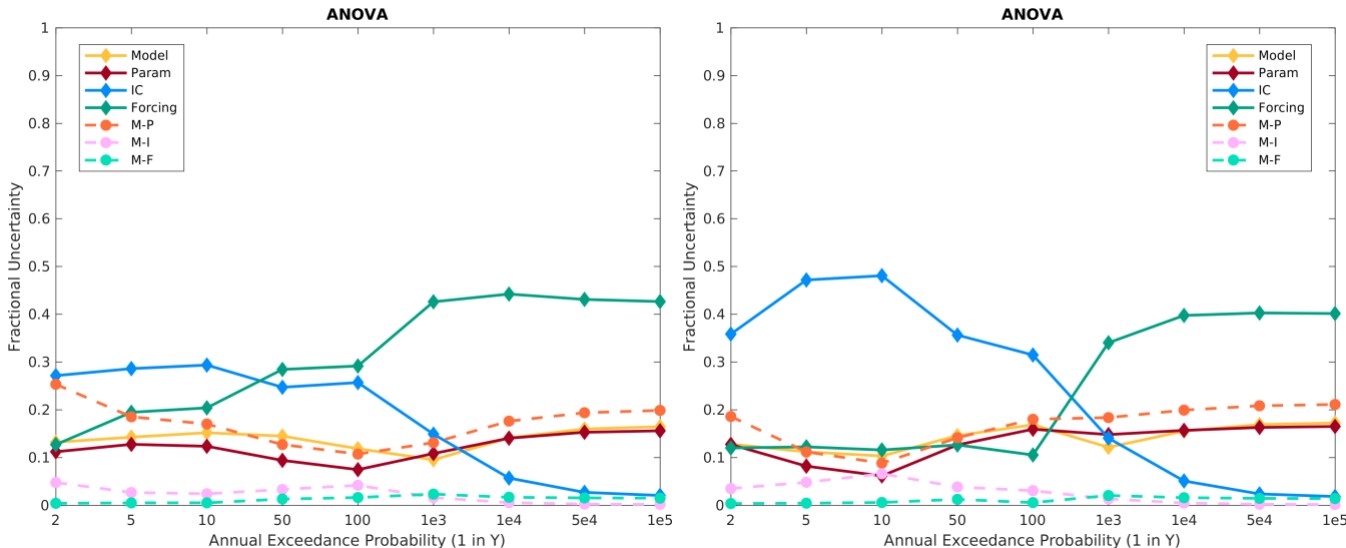

**Figure 9. Altus fractional variance contributions using all ten model structures for the a) dry meteorological sequence and b) historical meteorological sequence.**

730

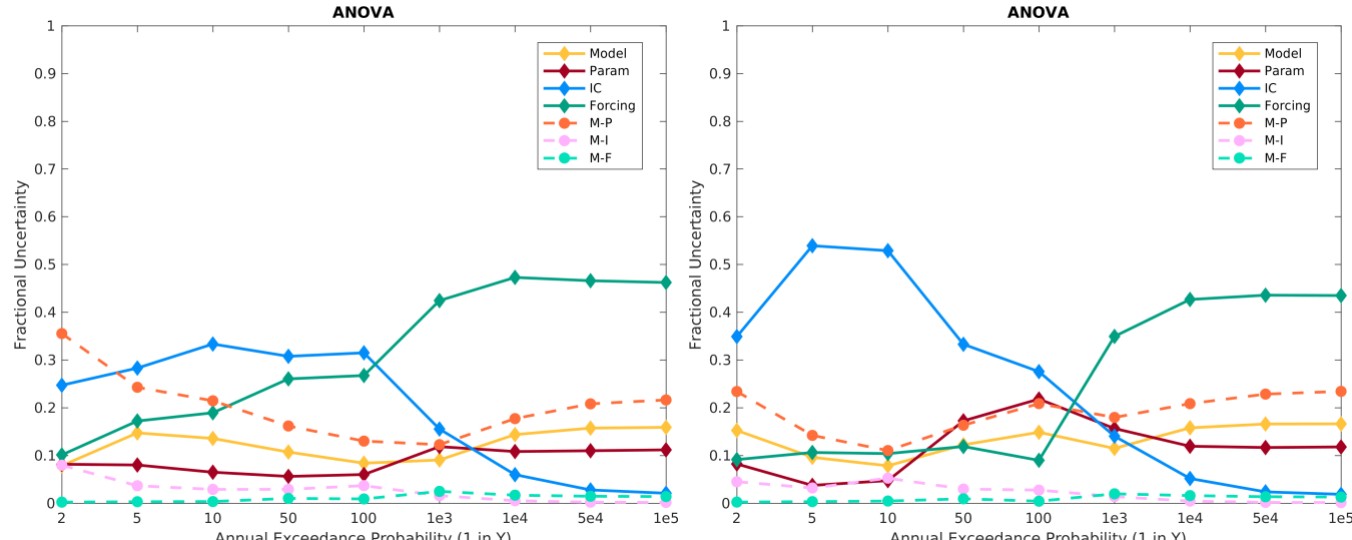

**Figure 10. Altus fractional variance contributions using the three base models: HEC-HMS, VIC, SAC-SMA, for the a) dry meteorological sequence and b) historical meteorological sequence.**

735

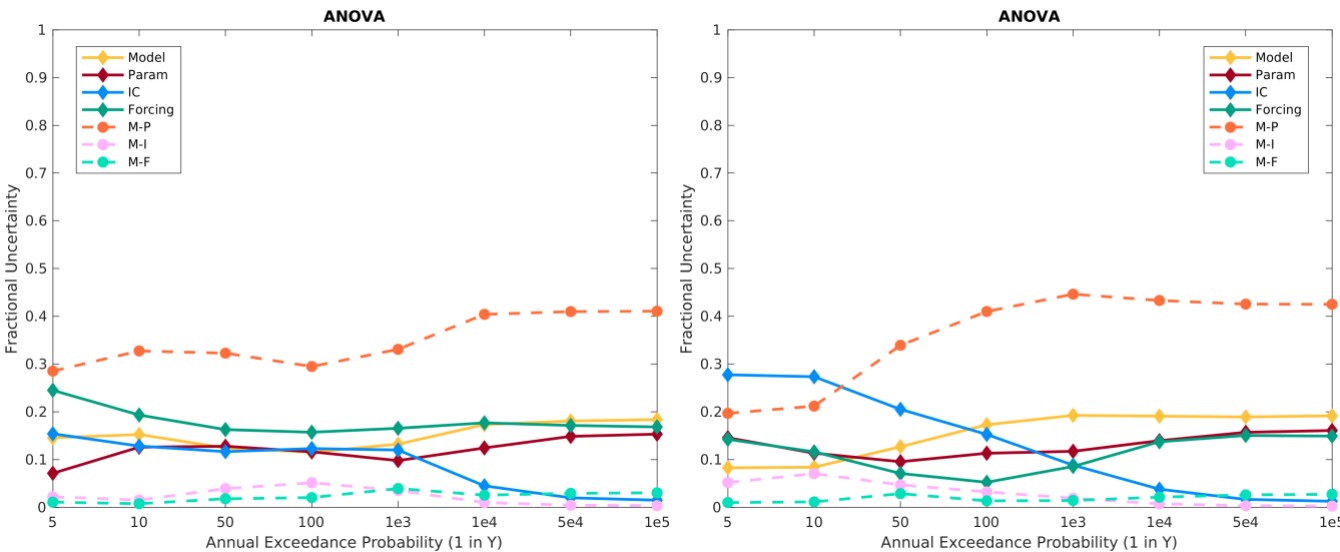

**Figure 11.** Altus fractional variance contributions for the two most disparate flood responses: SAC-SMA (Model #3), and a SAC-SMA/HEC-HMS combination (Model #6), for the a) dry metorological sequence and b) historical meteorological sequence.

740

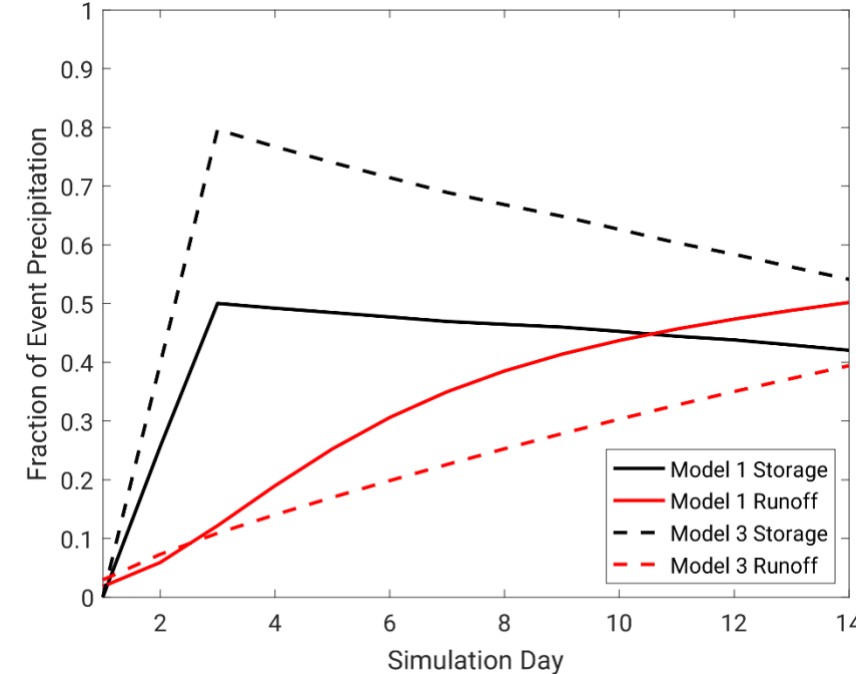

**Figure 12.** Change in storage (black lines) and cumulative runoff (red lines) normalized by flood event precipitation input for Model 1 (solid) and Model 3 (dashed) at Island Park for one precipitation frequency distribution bin using the median (50th percentile) precipitation frequency distribution.