# Peer review of "Identifying Sensitivities in Flood Frequency Analyses using a Stochastic Hydrologic Modeling System"

_Hydrology and Earth System Sciences, 2021_

## Author Comment (AC1)

Reply to Reviewer #1

We have replied to all the reviewer's comments in red.

The paper is well structured while the content is very dense and properly concise. The paper gives an important contribute to the analysis of factors affecting the FF curve by performing a deep analysis of the effects produced by models structure, model parameters, interaction between model parameters and model structure, initial conditions (in terms of water content) and precipitation events. In my opinion the paper can be accepted subject to minor revision.

We thank the reviewer for their positive assessment of the manuscript and their helpful specific comments highlighting areas for further improvement. Please see our point by point replies below.

I suggest the authors prepare a flux diagram describing their workflow, i.e. all the steps of their procedure. Indeed, there is a lot of attention on how the available data are used and this, in the end, slightly obscures the logic and the sequence of the steps. In this (or these) diagram(s) the authors should highlight the deep meaning of each step independently of the way the available data are used for their quantification.

We agree with your comment. After reexamining the paper, it is at points too concise and would benefit from at least one workflow diagram, further explanation, and improved terminology. We plan to add at least one detailed workflow diagram to the revised manuscript along with further text description highlighting the meaning of our methodological choices.

It is not completely clear to me how the rainfall events are generated starting from the precipitation frequency curve. I understand that simulations are performed with a time step of one day and thus precipitations are generated with this time step. However, in the case of Island Park, a two-day precipitation event is generated from the frequency curve and I do not understand if the total of precipitation in two days is generated or if the event is scanned at daily level.

We will add more explanation on this point. For Island Park, as you state, the precipitation frequency curve is a two day precipitation event total. Once we generate the precipitation events, we randomly split the events across two time steps so that the FUSE models receive precipitation over two days.

Overall, I prefer continuous simulations than event simulations: in fact, in this way the "natural" combination between rainfall periods and flood periods is obtained without any artificial combination between initial conditions and rainfall events. Instead of using the regional analysis, why don't the authors have set up a rainfall model (Neyman-Scott or Poisson model), performed long simulations and extracted annual maxima from them?

We agree that continuous simulations over many hundred to thousands of years is another way to perform FF estimation and has some possible benefits over event based modeling, although you still have to specify the rainfall model and all associated parameters as the reviewer notes. For this study the underlying motivation is to provide relevant results for the US Bureau of Reclamation, and event based

modeling is the method they use for their flood studies.  It would be interesting for someone to examine FF sensitivity in a similar framework to examine the methodological choices in the continuous modeling chain.

Incidentally, Hashimi -> Hashemi

Thank you for catching this typo, we apologize for the mistake.

---

## Author Comment (AC2)

Reply to Dr. Daniel Wright (Reviewer #2)

We have replied to all the reviewer's comments in red.

Review of HESS-2021-49: "Identifying Sensitivities in Flood Frequency Analyses using a Stochastic Hydrologic Modeling System" by Newman et al.

The authors present a sensitivity study examining the relatively contributions to flood quantiles from precipitation, initial conditions, model structure and parameters, and meteorological sequencing for two watersheds in the western US. The results are interesting and useful, while the manuscript is well-written. Like reviewer #1, I think that only minor revisions are needed. Similarly, like reviewer #1, I think occasionally found it difficult to understand exactly what was done or why. I'll point out those issues that I noticed, but I agree with reviewer #1 that an overall workflow diagram might be helpful if done with care.

Thank you Dan for the positive review and very helpful comments. We agree that the manuscript needs further clarification and an additional well formulated workflow figure. Please see our point by point replies below.

Specific comments:

I agree with the authors' discussion of AEP equality assumptions, but the potential problems don't end there. Even the assumption that precipitation annual maxima—which are the values used here and in most studies—are the drivers of streamflow annual maxima is not really correct. In Yu et al. (2019; specifically, Table 3 in that paper), we found that you need to get into 200+ year return periods before that assumption is really reliable, at least for the midsized midwestern watershed we looked at. Clearly, this is less of an issue for really big floods.

Thanks for the additional discussion related to AEP neutrality assumptions. This is an interesting point and we will certainly add some of this discussion and associated references into our introduction.

L68-70: It seems like some element is missing from this sentence. "higher sensitivity…" higher than what?

Thanks for catching this, we mean that some parameters related in the hydrologic model influence FF estimates more than others in a relative sense.

Section 3.1: I think you need to provide more explanation on how you used the total probability theorem. I *think* I understand what you did, but the reader shouldn't have to guess. Out of curiosity, I'm wondering if that approach would be valid when using distributed models. With lumped models (which I assume the authors are using here, but I'm not actually sure; see below), a bigger rainfall event combined with a higher IC will result in a higher peak than a smaller rainfall combined with a drier IC. But with distributed models, that is only true in general but not universally due to routing effects—I've seen cases where this isn't.

This general point was also highlighted by the other two reviewers. Thus, we will improve our discussion regarding our use of the total probability theorem in section 3.1, and possibly add more detail to other methodological points in this section.

Also, thanks for the ideas regarding lumped versus distributed models. We are indeed using lumped models in this study. We will consider adding a few sentences in the introduction and possibly in this section regarding the nuances between lumped and distributed models and how that may impact an FF sensitivity analysis like ours.

Section 3.4: It would have been nice to know how important this assumption of picking a few (high) ICs is, as opposed to letting the ICs vary more widely. My particular concern is that to some degree or another, your rainfall quantiles are probably based in part on some events (probably some big ones!) that are outside of this Feb-July (Altus) and Apr-June (Island Park) periods. The Colorado 2013 floods are a good example of this. I suspect that there is some degree of misrepresentation of the relative importance of ICs and precipitation for this reason.

This is a good question, and one we will add some general discussion about in the manuscript. For the current study, we chose to focus on wetter initial conditions (ICs) to focus on floods with large return periods (e.g. 10,000+ years) and following general Reclamation FF estimation methodologies. Admittedly, we are showing results across even frequent return periods and the reliance on only wet ICs may influence the importance of IC uncertainty for these more frequent return periods, also mentioned by reviewer #3.

In general Reclamation focuses on larger events and wetter periods of the observed record and generally uses the range of conditions for those larger events to inform the distribution of initial conditions sampling from those events. This assumption may not be valid in all hydrologic regimes or for all events, especially in more arid environments and for unique events like the Colorado 2013 floods.

We plan to add the above clarification and discussion and emphasize the point that there may be an underestimation of the sensitivity of ICs, particularly for frequent return periods.

Section 3.5: I found the explanation of spatial precipitation structure to be unclear-both how it was done, and why it was done. In the latter case, my confusion stems from the lack of description of the models' spatial discretization (or maybe I missed that somewhere).

We will clarify that the model is lumped, therefore there is no spatial structure to the precipitation inputs. Using the gridded distributed precipitation inputs, we combine all grid cells that intersect the basin polygon using their fractional areas to create the basin mean precipitation input.

Section 3.6: I found this section difficult to follow, and didn't totally understand what was being done.

We agree this section is poorly worded and may be better served by including an example figure of the two different precipitation sequences. Essentially, we are trying to distinguish between using the standard Reclamation approach of event modeling where they have the event precipitation input

followed by zero precipitation for the rest of the simulation, and a more realistic approach of specifying the event precipitation and then continuing with historical precipitation after the specified event.

Section 3.7: While neither Peleg et al. (2017) for I (in Zhu et al. 2018) examined model structure, we did use ANOVA (in Zhu et al.) or something like ANOVA (in Peleg et al.) to examine the roles of other things (ICs, for one) in FFA. I won't be offended if you don't, but you may consider whether those prior studies' findings provide relevant contrasts with your work.

Thank you for these citations.  We will examine these two studies and include relevant discussion of them in the introduction, this section, and throughout the results.

L304: Usage of "overrepresentation" is unclear.

Thanks for catching this, we meant to say 'overestimation'.

L304 and more generally: given all the moving parts here, some section referencing would help, as well as a bit more precision with terminology. For example, "KGE interval metric-based calibration"-it took me a minute to figure out what you were talking about. You mean calibration based on peak flows, right? Furthermore, referring back to Section 3.3 (e.g. "(see Section 3.3)") would help the reader the track down the relevant details they might have missed or forgotten. This section referencing would help in a number of other places too.

Thanks for the helpful suggestion.  We agree that section referencing may help improve detail tracking for readers.  We will identify places in the text where we can do this and use the subsequent comments from your review as well.  We will also work on our terminology to improve precision and readability.

You are correct that 'KGE interval metric-based calibration' refers to annual peak flow calibration using KGE.

Section 5: It would be nice to know if the "shapes" of the flood frequency curves are driven by the shapes of the precip frequency curves, which aren't shown.

We will add a figure highlighting the precipitation frequency distributions.  In general the shapes of the flood frequency curves do roughly follow the precipitation frequency curve shapes.

L365 and around there: I struggled with this paragraph, in part because I didn't understand the descriptions in Section 3.6. Also, this is another good place to refer back to earlier sections/descriptions.

Thanks for the suggestion.  We will modify this paragraph and add section referencing.

L376: "Dry to historical meteorological sequence"-I found this wording confusing

We will modify this phase within the sentence.

L390: You could refer back to the first mention that you're analyzing different streamflow timescales

We will add reference to the correct previous section.

L428: consider replacing "across" with "between"

We will change 'across' to 'between'.

References:

Yu, G., D. B. Wright, Z. Zhu, C. Smith, and K. D. Holman. "Process-Based Flood Frequency Analysis in an Agricultural Watershed Exhibiting Nonstationary Flood Seasonality." Hydrol. Earth Syst. Sci. 23, no. 5 (May 7, 2019): 2225–43. https://doi.org/10.5194/hess-23-2225-2019.

Zhu, Zhihua, Daniel B. Wright, and Guo Yu. "The Impact of Rainfall Spaceâ Time Structure in Flood Frequency Analysis." Water Resources Research 54, no. 11 (2018): 8983–98. https://doi.org/10.1029/2018WR023550.

Peleg, N., F. Blumensaat, P. Molnar, S. Fatichi, and P. Burlando. "Partitioning the Impacts of Spatial and Climatological Rainfall Variability in Urban Drainage Modeling." Hydrol. Earth Syst. Sci. 21, no. 3 (March 14, 2017): 1559–72. https://doi.org/10.5194/hess-21-1559-2017.

---

## Author Comment (AC3)

Reply to Reviewer #3

We have replied to all the reviewer's comments in red.

General comments :

This paper presents a comprehensive study of the uncertainties of a Flood Frequency analysis method based on stochastic simulation. The different sources of uncertainty are distributed between the structure of the model, the estimation of the parameters, the initial conditions and the inputs (rainfall). Overall, this paper is well written and presents significant and comprehensive scientific results. As the other reviewers I will recommend minor corrections, mainly to clarify some points about the tools used, described only by publications. Clarifications on the methodology would allow a better understanding of certain points detailed below.

We thank the reviewer for their positive assessment of the manuscript, and their helpful specific comments highlighting areas for further improvement. Please see our point by point replies below.

Specific comments :

Section 3.1: As the other reviewers, I think that a diagram presenting the workflow would help in understanding the different steps and tools used.

Thank you for the comment, we agree with the need for further clarification through development of at least one workflow diagram and additional text describing our methods and tools.

L134: specify if the modeling is lumped or distributed, knowing that the input data for the calibration looks distributed (in Newman et al, 2015). Clarify how you calibrate the hydrological models from an ensemble of historical meteorology (is the ensemble related to ground data interpolation uncertainties?). If the modeling is global, are the ensemble really very different sets (spatial mean)?

For this study the model is lumped. We will clarify this around line 134. We combine all grid cells that intersect the basin polygon using their fractional areas to create the basin mean meteorology. We calibrate each ensemble member to arrive at 100 model parameter sets per basin.

The ensemble estimates interpolation and instrument uncertainty as described in Clark and Slater (2006) and Newman et al. (2015). We will add a brief description of this methodology and dataset to the revised paper.

L137: Clarify if "two event sequence possibilities" are in fact two periods/seasons? And which ones?

We will clarify this phrase with the following: The two event sequences are 1) Randomly selected historical precipitation and temperature for the days following the specified event precipitation; and 2) the same historical temperature time series but with zero post-event precipitation.

L141: We ask ourselves the question of 11 parameters sets: how are they obtained ? This is not from the sets of historical meteorology since there are 100 used. (ok, the answer is in line 328... but only 10 sets for the Altus basin)

Thank you for catching this omission, we will add an explanation of the model parameter set selection earlier in the paper.

L220: why not take the whole distribution of initial conditions (IC) and only the strongest initial conditions. This does not allow to associate dry CI with heavy rainfall, and it reduces the impact of the uncertainties related to IC (for the current frequencies at least).

This is a good question, and one we will add some general discussion about in the manuscript. For the current study, we chose to focus on wetter initial conditions (ICs) to focus on floods with large return periods (e.g. 10,000+ years) and following general Reclamation FF estimation methodologies. Admittedly, we are showing results across even frequent return periods and the reliance on only wet ICs may influence the importance of IC uncertainty for these more frequent return periods as mentioned by Reviewer #2.

In general Reclamation focuses on larger events and wetter periods of the observed record and generally uses the range of conditions for those larger events to inform the distribution of initial conditions sampling from those events. This assumption may not be valid in all hydrologic regimes, especially in more arid environments where conditions such as surface sealing and rock-mantled slopes may actually result in more severe flooding under intense short-duration thunderstorms. While the basins tested here did not consider those conditions, we agree that in the appropriate hydrologic regime, users should consider a wider distribution for such initial conditions which may increase the importance of initial conditions in flood response.

We plan to add the above clarification and discussion and emphasize the point that there may be an underestimation of the importance of ICs, particularly for frequent return periods.

L250: Rainfall events are in fact total rainfall generated from a regional probability law. Is it a limitation of the method, to simulate two-day events to generate extreme flood flows? As the time step of the hydrological modeling is daily, how are the rains distributed over the two days simulated? Moreover, can you explain how the problem of changing from a point rainfall to a basin rainfall is solved (how the areal-reduction factor is taken into account).

The precipitation frequency distribution is a two-day total only for Island Park, for Altus it is a single day event.  We will be sure to clarify this point.  For time splitting in Island Park, we randomly split the events across two days so that the FUSE models receive the precipitation over two days.

We will also add additional clarification regarding how we applied the areal-reduction factor (ARF) for these specific basins.  In general, the process of converting a point precipitation-frequency curve (which is generally considered valid over 10 mi$^2$) to a basin-average precipitation-frequency curve can be

accomplished through the application of an ARF.  According to Reclamation (2012), the authors developed a linear relationship between point (x-axis) and basin-average (y-axis) storm totals from 12 different storms that impacted the Altus Dam region identified in HMR 51(Schreiner and Riedel, 1978), HMR 52 (Hansen et al., 1982), and HMR 55A (Hansen et al., 1988).

Although some published studies in the literature demonstrate that ARFs vary as a function of annual exceedance probability for common to extreme events (e.g., Bell, 1976), authors of the Island Park study applied a constant ARF for all AEPs.  More specifically, Reclamation (2015) multiplied the point-specific precipitation-frequency curve by a constant ARF of 0.85, which they estimated using historical point and basin-average precipitation depths available in HMR 55A(Hansen et al. 1988) and HMR 57 (Hansen et. al.1994).

L250-252 and 260-262: it should be better explained how you use historical precipitation events that are equal to the basin-average magnitudes sampled from the frequency curve, especially for extremes events.

Thank you for this comment.  To clarify, the sampled and simulated precipitation events that are equal to the basin-average magnitude sampled from the frequency curve would be scaled by a factor of 1 (i.e., no change).  Sampled and simulated precipitation events that are greater than the sampled basin-average magnitude sampled from the frequency curve would be scaled by a factor less than 1 (i.e., decreased). Finally, sampled and simulated precipitation events that are less than the sampled basin-average magnitude sampled from the frequency curve would be scaled by a factor greater than 1 (i.e., increased).

L309: How is the calibration performance assessed. There do not appear to be any validation procedures to verify this (as LOO procedure for instance). The graphs in Figure 4 presented calibration results. It is therefore difficult to judge the performance of the different models, which also depends on their robustness. It is better to show validation results.

Thank you for bringing this point up.  We agree that out of sample validation results, particularly for a forecast application, are the appropriate way to evaluate model performance.  In this case we had two factors  to consider for how to calibrate and validate our models.  First, we had limited data available in both catchments for calibration, and second we were not performing a true 'forecast' application in the sense of worrying about absolute values for floods.  Thus, we decided to use all available data for calibration and select models using the in-sample performance.  Going forward this would be an area for exploration to identify split sample calibration-validation strategies for floods to minimize model behavioral changes, such as changes in the model sensitivity explored here in the validation phase.

Specific comments : Figure 1: as it is a theoretical explanation diagram, put theoretical curves (straight lines for example ?) because we have the impression that it is a result.

Thank you for the figure suggestion.  We will modify Figure 1 by simplifying the curves to be much smoother, such as exponential or square root functions, but may not move to straight lines.

**References:**

Bell, F.C. 1976. The areal reduction factors in rainfall-frequency estimation. Natural Environmental Research Council, Report 35. Institute of Hydrology, Wallingford, United Kingdom.

Clark, M. P., and A. Slater 2006: Probabilistic quantitative precipitation estimation in complex terrain. J. Hydrometeor., **7**, 3–22, doi:10.1175/JHM474.1.

Hansen, E.M., Schreiner, L.C., and Miller, J.F. (1982) Application of Probable Maximum Precipitation Estimates, United States East of the 105th Meridian. Hydrometeorological Report No. 52, National Weather Service, National Oceanic and Atmospheric Administration, U.S. Department of Commerce, Silver Spring, MD, 168 p.

Hansen, E.M., Fenn, D.D., Schreiner, L.C., Stodt, R.W., and Miller, J.F. (1988) Probable Maximum Precipitation Estimates-United States between the Continental Divide and the 103rd Meridian. Hydrometeorological Report No. 55A, National Weather Service, National Oceanic and Atmospheric Administration, U.S. Department of Commerce, Silver Spring, MD, 242 p.

Hansen, E.M., Fenn, D.D., Corrigan, P., Vogel, J.L., Schreiner, L.C. and Stodt, R.W. (1994) Probable Maximum Precipitation-Pacific Northwest States, Columbia River (including portions of Canada), Snake River and Pacific Coastal Drainages. Hydrometeorological Report No. 57, National Weather Service, National Oceanic and Atmospheric Administration, U.S. Department of Commerce, Silver Spring, MD, 338 p.

Newman, A. J., Clark, M. P., Craig, J., Nijssen, B., Wood, A., Gutmann, E., Mizukami, N., Brekke, L. D., and Arnold, J. R. (2015). Gridded ensemble precipitation and temperature estimates for the contiguous United States. Journal of Hydrometeorology, 16(6), 2481-2500.

Schreiner, L.C. and Riedel, J.T. (1978) Probable Maximum Precipitation Estimates, United States East of the 105th Meridian. Hydrometeorological Report No. 51, National Weather Service, National Oceanic and Atmospheric Administration, U.S. Department of Commerce, Silver Spring, MD, 87 p.

---

## Author Response (AR1)

Reply to Reviewer #1

We have replied to all the reviewer's comments in red.

The paper is well structured while the content is very dense and properly concise. The paper gives an important contribute to the analysis of factors affecting the FF curve by performing a deep analysis of the effects produced by models structure, model parameters, interaction between model parameters and model structure, initial conditions (in terms of water content) and precipitation events. In my opinion the paper can be accepted subject to minor revision.

We thank the reviewer for their positive assessment of the manuscript and their helpful specific comments highlighting areas for further improvement. Please see our point-by-point replies below.

I suggest the authors prepare a flux diagram describing their workflow, i.e. all the steps of their procedure. Indeed, there is a lot of attention on how the available data are used and this, in the end, slightly obscures the logic and the sequence of the steps. In this (or these) diagram(s) the authors should highlight the deep meaning of each step independently of the way the available data are used for their quantification.

We have added a workflow diagram as new Figure 3. Sections 3.1, 3.4, and 3.6 have been extensively edited, and additional clarification has been added to Section 3.7.

It is not completely clear to me how the rainfall events are generated starting from the precipitation frequency curve. I understand that simulations are performed with a time step of one day and thus precipitations are generated with this time step. However, in the case of Island Park, a two-day precipitation event is generated from the frequency curve and I do not understand if the total of precipitation in two days is generated or if the event is scanned at daily level.

This is clarified in Figure 3 and Section 3.1.

Overall, I prefer continuous simulations than event simulations: in fact, in this way the "natural" combination between rainfall periods and flood periods is obtained without any artificial combination between initial conditions and rainfall events. Instead of using the regional analysis, why don't the authors have set up a rainfall model (Neyman-Scott or Poisson model), performed long simulations and extracted annual maxima from them?

We agree that continuous simulations over many hundreds to thousands of years is another way to perform FF estimation We have added discussion of this point in Section 3.1.

Incidentally, Hashimi -> Hashemi

Thank you for catching this typo, we have corrected all instances of it.

Reply to Dr. Daniel Wright (Reviewer #2)

We have replied to all the reviewer's comments in red.

Review of HESS-2021-49: "Identifying Sensitivities in Flood Frequency Analyses using a Stochastic Hydrologic Modeling System" by Newman et al.

The authors present a sensitivity study examining the relatively contributions to flood quantiles from precipitation, initial conditions, model structure and parameters, and meteorological sequencing for two watersheds in the western US. The results are interesting and useful, while the manuscript is well-written. Like reviewer #1, I think that only minor revisions are needed. Similarly, like reviewer #1, I think occasionally found it difficult to understand exactly what was done or why. I'll point out those issues that I noticed, but I agree with reviewer #1 that an overall workflow diagram might be helpful if done with care.

Thank you, Dan, for the positive review and very helpful comments. We agree that the manuscript needs further clarification and an additional well formulated workflow figure. Please see our point by point replies below.

Specific comments:
I agree with the authors' discussion of AEP equality assumptions, but the potential problems don't end there. Even the assumption that precipitation annual maxima—which are the values used here and in most studies—are the drivers of streamflow annual maxima is not really correct. In Yu et al. (2019; specifically, Table 3 in that paper), we found that you need to get into 200+ year return periods before that assumption is really reliable, at least for the midsized midwestern watershed we looked at. Clearly, this is less of an issue for really big floods.

Thanks for the additional discussion related to AEP neutrality assumptions. This is an interesting point and we have added brief mention in the introduction on lines 97-99.

L68-70: It seems like some element is missing from this sentence. "higher sensitivity…" higher than what?

We have corrected this sentence.

Section 3.1: I think you need to provide more explanation on how you used the total probability theorem. I *think* I understand what you did, but the reader shouldn't have to guess. Out of curiosity, I'm wondering if that approach would be valid when using distributed models. With lumped models (which I assume the authors are using here, but I'm not actually sure; see below), a bigger rainfall event combined with a higher IC will result in a higher peak than a smaller rainfall combined with a drier IC. But with distributed models, that is only true in general but not universally due to routing effects—I've seen cases where this isn't.

We have rewritten most of section 3.1 (lines 175-200) and added a detailed workflow diagram as new Figure 3. We have clarified that we are using a watershed model in this study on lines 183 and 256.

Section 3.4: It would have been nice to know how important this assumption of picking a few (high) ICs is, as opposed to letting the ICs vary more widely. My particular concern is that to some degree or

another, your rainfall quantiles are probably based in part on some events (probably some big ones!) that are outside of this Feb-July (Altus) and Apr-June (Island Park) periods. The Colorado 2013 floods are a good example of this. I suspect that there is some degree of misrepresentation of the relative importance of ICs and precipitation for this reason.

We have added discussion of this point in section 3.4 on lines 309-316.

Section 3.5: I found the explanation of spatial precipitation structure to be unclear-both how it was done, and why it was done. In the latter case, my confusion stems from the lack of description of the models' spatial discretization (or maybe I missed that somewhere).

We have clarified that we are using a watershed (lumped) model in this study on lines 183 and 256.

Section 3.6: I found this section difficult to follow, and didn't totally understand what was being done.

We have shortened and clarified Section 3.6.

Section 3.7: While neither Peleg et al. (2017) for I (in Zhu et al. 2018) examined model structure, we did use ANOVA (in Zhu et al.) or something like ANOVA (in Peleg et al.) to examine the roles of other things (ICs, for one) in FFA. I won't be offended if you don't, but you may consider whether those prior studies' findings provide relevant contrasts with your work.

Thank you for these citations. We will examine these two studies and include relevant discussion of them in the introduction, this section, and throughout the results.

L304: Usage of "overrepresentation" is unclear.

We meant to say 'overestimation'.

L304 and more generally: given all the moving parts here, some section referencing would help, as well as a bit more precision with terminology. For example, "KGE interval metric-based calibration"-it took me a minute to figure out what you were talking about. You mean calibration based on peak flows, right? Furthermore, referring back to Section 3.3 (e.g. "(see Section 3.3)") would help the reader the track down the relevant details they might have missed or forgotten. This section referencing would help in a number of other places too.

We have improved our wording – e.g. "KGE interval metric-based calibration" is now annual peak flow calibration, and added references to specific sections throughout the paper.

Section 5: It would be nice to know if the "shapes" of the flood frequency curves are driven by the shapes of the precip frequency curves, which aren't shown.

We have added new Figure 4 to show the precipitation frequency curves.

L365 and around there: I struggled with this paragraph, in part because I didn't understand the descriptions in Section 3.6. Also, this is another good place to refer back to earlier sections/descriptions.

We added section referencing here.

L376: "Dry to historical meteorological sequence"-I found this wording confusing

We have modified the discussion of the meteorology used during the event simulations in the re-written Section 3.6 and throughout the discussion in Section 5.

L390: You could refer back to the first mention that you're analyzing different streamflow timescales

We have added to this.

L428: consider replacing "across" with "between"

Changed.

References:
Yu, G., D. B. Wright, Z. Zhu, C. Smith, and K. D. Holman. "Process-Based Flood Frequency Analysis in an Agricultural Watershed Exhibiting Nonstationary Flood Seasonality." Hydrol. Earth Syst. Sci. 23, no. 5 (May 7, 2019): 2225–43. https://doi.org/10.5194/hess-23-2225-2019.

Zhu, Zhihua, Daniel B. Wright, and Guo Yu. "The Impact of Rainfall Space     Time Structure in Flood Frequency Analysis." Water Resources Research 54, no. 11 (2018): 8983–98. https://doi.org/10.1029/2018WR023550.

Peleg, N., F. Blumensaat, P. Molnar, S. Fatichi, and P. Burlando. "Partitioning the Impacts of Spatial and Climatological Rainfall Variability in Urban Drainage Modeling." Hydrol. Earth Syst. Sci. 21, no. 3 (March 14, 2017): 1559–72. https://doi.org/10.5194/hess-21-1559-2017.

Reply to Reviewer #3

We have replied to all the reviewer's comments in red.

General comments:
This paper presents a comprehensive study of the uncertainties of a Flood Frequency analysis method based on stochastic simulation. The different sources of uncertainty are distributed between the structure of the model, the estimation of the parameters, the initial conditions and the inputs (rainfall). Overall, this paper is well written and presents significant and comprehensive scientific results. As the other reviewers I will recommend minor corrections, mainly to clarify some points about the tools used, described only by publications. Clarifications on the methodology would allow a better understanding of certain points detailed below.

We thank the reviewer for their positive assessment of the manuscript, and their helpful specific comments highlighting areas for further improvement. Please see our point-by-point replies below.

Specific comments:
Section 3.1: As the other reviewers, I think that a diagram presenting the workflow would help in understanding the different steps and tools used.

Thank you for the comment, see our replies to reviewers #1 and #2 regarding improving clarification and the new Figure 3.

L134: specify if the modeling is lumped or distributed, knowing that the input data for the calibration looks distributed (in Newman et al, 2015). Clarify how you calibrate the hydrological models from an ensemble of historical meteorology (is the ensemble related to ground data interpolation uncertainties?). If the modeling is global, are the ensemble really very different sets (spatial mean)?

For this study the model is lumped. We have clarified this on lines 183 and 256.

L137: Clarify if "two event sequence possibilities" are in fact two periods/seasons? And which ones?

We have clarified Section 3.6 and reworded our description of our two different meteorology inputs for our two sets of event-based modeling.

L141: We ask ourselves the question of 11 parameters sets: how are they obtained? This is not from the sets of historical meteorology since there are 100 used. (ok, the answer is in line 328... but only 10 sets for the Altus basin)

We have clarified this on lines 214-216.

L220: why not take the whole distribution of initial conditions (IC) and only the strongest initial conditions. This does not allow to associate dry CI with heavy rainfall, and it reduces the impact of the uncertainties related to IC (for the current frequencies at least).

We have added discussion of this in Section 3.4.

L250: Rainfall events are in fact total rainfall generated from a regional probability law. Is it a limitation of the method, to simulate two-day events to generate extreme flood flows? As the time step of the hydrological modeling is daily, how are the rains distributed over the two days simulated? Moreover, can you explain how the problem of changing from a point rainfall to a basin rainfall is solved (how the areal-reduction factor is taken into account).

The precipitation frequency distribution is a two-day total only for Island Park, for Altus it is a single day event. For time splitting in Island Park, we randomly split the events across two days so that the FUSE models receive the precipitation over two days. These points are clarified in Figure 3 and Section 3.1.

We have added additional clarification regarding how we applied the areal-reduction factor (ARF) for these specific basins in Section 3.5.

L250-252 and 260-262: it should be better explained how you use historical precipitation events that are equal to the basin-average magnitudes sampled from the frequency curve, especially for extremes events.

We have clarified how precipitation events are generated from the frequency curves in Section 3.5 and clarified how we are using them in event simulations in Sections 3.6 and 3.7.

L309: How is the calibration performance assessed. There do not appear to be any validation procedures to verify this (as LOO procedure for instance). The graphs in Figure 4 presented calibration results. It is therefore difficult to judge the performance of the different models, which also depends on their robustness. It is better to show validation results.

We have used all data for calibration and have not performed an out-of-sample evaluation of the model as we are not concerned with true forecast performance and need the largest possible sample sizes for calibration. This is clarified on lines 295-297.

Specific comments:

Figure 1: as it is a theoretical explanation diagram, put theoretical curves (straight lines for example?) because we have the impression that it is a result.

Thank you for the figure suggestion. We have chosen to leave Figure 1 as is.

**References:**
Bell, F.C. 1976. The areal reduction factors in rainfall-frequency estimation. Natural Environmental Research Council, Report 35. Institute of Hydrology, Wallingford, United Kingdom.

Clark, M. P., and A. Slater 2006: Probabilistic quantitative precipitation estimation in complex terrain. J. Hydrometeor., 7, 3–22, doi:10.1175/JHM474.1.

Hansen, E.M., Schreiner, L.C., and Miller, J.F. (1982) Application of Probable Maximum Precipitation Estimates, United States East of the 105th Meridian. Hydrometeorological Report No. 52, National Weather Service, National Oceanic and Atmospheric Administration, U.S. Department of Commerce, Silver Spring, MD, 168 p.

Hansen, E.M., Fenn, D.D., Schreiner, L.C., Stodt, R.W., and Miller, J.F. (1988) Probable Maximum Precipitation Estimates-United States between the Continental Divide and the 103rd Meridian. Hydrometeorological Report No. 55A, National Weather Service, National Oceanic and Atmospheric Administration, U.S. Department of Commerce, Silver Spring, MD, 242 p.

Hansen, E.M., Fenn, D.D., Corrigan, P., Vogel, J.L., Schreiner, L.C. and Stodt, R.W. (1994) Probable Maximum Precipitation-Pacific Northwest States, Columbia River (including portions of Canada), Snake River and Pacific Coastal Drainages. Hydrometeorological Report No. 57, National Weather Service, National Oceanic and Atmospheric Administration, U.S. Department of Commerce, Silver Spring, MD, 338 p.

Newman, A. J., Clark, M. P., Craig, J., Nijssen, B., Wood, A., Gutmann, E., Mizukami, N., Brekke, L. D., and Arnold, J. R. (2015). Gridded ensemble precipitation and temperature estimates for the contiguous United States. Journal of Hydrometeorology, 16(6), 2481-2500.

Schreiner, L.C. and Riedel, J.T. (1978) Probable Maximum Precipitation Estimates, United States East of the 105th Meridian. Hydrometeorological Report No. 51, National Weather Service, National Oceanic and Atmospheric Administration, U.S. Department of Commerce, Silver Spring, MD, 87 p.

---

## Author Response (AR2)

Author Response:

Comments from Dan Wright are in black, our responses are in red.

The authors have addressed nearly all of my concerns. One minor remaining issue that should be "cleaned up" is that the description of how the total probability theorem is used appears to be found in Section 3.1 lines 145-149. However, Section 3.1 line 133 seems to imply that it is described instead in Section 3.6. Section 3.6 does not appear to contain any information regarding this step. This inconsistency should be corrected, and the authors should make explicit where (within the manuscript) and how this theorem is used.

We thank Dan for catching this technical error. We have clarified line 133 to note the total probability theorem is mentioned in Section 3.1, lines 145-149, and then described in more detail in Section 3.7 (not 3.6) in the first paragraph.